# Representation of genomic intratumor heterogeneity in multi-region non-small cell lung cancer patient-derived xenograft models

Robert E. Hynds [1,2,3,108] ✉, Ariana Huebner [1,2,4,108], David R. Pearce [1,2,108], Mark S. Hill [2], Ayse U. Akarca [5], David A. Moore [1,2,5], Sophia Ward [1,2,6], Kate H. C. Gowers[7], Takahiro Karasaki [1,2,8], Maise Al Bakir [1,2], Gareth A. Wilson[2], Oriol Pich [1,2], Carlos Martínez-Ruiz[1,4], A. S. Md Mukarram Hossain [9,10], Simon P. Pearce [9,10], Monica Sivakumar[1,5], Assma Ben Aissa[11], Eva Grönroos [2], Deepak Chandrasekharan [7], Krishna K. Kolluri[7], Rebecca Towns[12], Kaiwen Wang[13], Daniel E. Cook [2], Leticia Bosshard-Carter[1,7], Cristina Naceur-Lombardelli [1], Andrew J. Rowan[2], Selvaraju Veeriah[1], Kevin Litchfield [1,14], Philip A. J. Crosbie[10,15], Caroline Dive [9,10], Sergio A. Quezada [1,11], Sam M. Janes [7], Mariam Jamal-Hanjani [1,8,16], Teresa Marafioti[5], TRACERx consortium*, Nicholas McGranahan [1,4] ✉ & Charles Swanton [1,2,16] ✉

Patient-derived xenograft (PDX) models are widely used in cancer research. To investigate the genomic fidelity of non-small cell lung cancer PDX models, we established 48 PDX models from 22 patients enrolled in the TRACERx study. Multi-region tumor sampling increased successful PDX engraftment and most models were histologically similar to their parent tumor. Whole-exome sequencing enabled comparison of tumors and PDX models and we provide an adapted mouse reference genome for improved removal of NOD *scid* gamma (NSG) mouse-derived reads from sequencing data. PDX model establishment caused a genomic bottleneck, with models often representing a single tumor subclone. While distinct tumor subclones were represented in independent models from the same tumor, individual PDX models did not fully recapitulate intratumor heterogeneity. On-going genomic evolution in mice contributed modestly to the genomic distance between tumors and PDX models. Our study highlights the importance of considering primary tumor heterogeneity when using PDX models and emphasizes the benefit of comprehensive tumor sampling.

In patient-derived xenograft (PDX) models, human tumors are propagated by transplantation into immunodeficient mice[1]. PDX models have become important models in cancer biology: they are thought to mimic tumor biology more closely than traditional cell lines as a consequence of their in vivo cell-cell and/or cell-matrix interactions, 3D architecture and relatively recent derivation[2]. Many reports have suggested that the drug responses of PDX models are concordant with those observed in either individual patients or in patient cohorts.

A full list of affiliations appears at the end of the paper. *A list of authors and their affiliations appears at the end of the paper. ✉e-mail: rob.hynds@ucl.ac.uk; nicholas.mcgranahan.10@ucl.ac.uk; charles.swanton@crick.ac.uk

This has led to the use of PDX models in pre-clinical drug trials prior to patient investigations[3], and to enthusiasm for the use of PDX models in personalized medicine approaches in which they are used as 'avatars' for individual patient responses to therapy in 'co-clinical' trials[4,5].

For pre-clinical oncology applications, the fidelity of PDX models is of major importance. Across cancer types, including non-small cell lung cancer (NSCLC)[6], PDX models bear histological similarity to the tumors from which they were derived. However, recent high-resolution analyses of breast cancer PDX models suggest that PDX models, like patient tumors, can comprise multiple genetically defined subclones[7] and that these undergo dynamic changes in their relative abundance during PDX engraftment and expansion[8]. Analysis of PDX model copy number profiles has cast doubt upon their representation of tumor molecular heterogeneity, specifically with regard to genomic evolution within the mouse[9,10]. While some of these differences may be attributable to technical issues surrounding the estimation of copy number profiles from RNA sequencing data, disagreement about the extent and importance of PDX copy number divergence remains when considering DNA sequencing data[10,11]. While some studies have included examples of matched patient-PDX pairs or the derivation of multiple PDX models from the same tumor, the genomic evolution during PDX model establishment and propagation has not been systematically assessed. Furthermore, the role of spatial sampling has not been explored and studies to date have not been performed in the context of multi-region patient sequencing data to formally establish how well PDX models represent the complex subclonal nature of primary tumors and their metastases.

Lung TRACERx is a prospective cohort study that aims to characterize the evolutionary dynamics of NSCLC through a multi-region whole-exome sequencing (WES) approach[12]. Here, we derive PDX models from multiple regions of primary NSCLC from patients enrolled in the TRACERx study to determine the histological and genetic fidelity of the PDX approach. By comparing WES data from initial passage zero (P0; i.e. the first xenograft tumor) PDX models, established passage three (P3) PDX models and multiply-sampled matched primary tumors, we investigate key unresolved issues in the use of PDX models. These include the extent of genomic bottlenecking upon engraftment, the reproducibility of PDX derivation across spatially distinct replicate samples and the emergence of de novo genetic alterations in PDX models during their propagation in mice. Further, we highlight the utility of a host-matched NSG-adapted reference genome to deconvolve human and mouse sequencing reads from PDX models.

## Results

### Establishment of PDX models from multiply-sampled NSCLC tumors

Primary non-small cell lung cancers (NSCLCs) from patients enrolled in the lung TRACERx study undergo multi-region whole-exome sequencing (WES) using a defined sampling protocol[12]. To characterize tumor evolution during patient-derived xenograft (PDX) model engraftment and propagation, we obtained tumor region-matched tissue and created PDX models from a representative patient subset (Fig. 1A; Supplementary Fig. 1). 145 specimens from 44 patients undergoing surgical resection of their primary NSCLC were injected subcutaneously in NOD *scid* gamma (NSG) mice, generating 64 xenografts from a cohort with diagnoses of lung adenocarcinoma (LUAD), lung squamous cell carcinoma (LUSC) and other NSCLC histological subtypes (including adenosquamous carcinoma, a collision tumor containing both LUSC and LUAD, combined LUAD and small cell carcinoma, pleomorphic carcinoma, carcinosarcoma, and large cell neuroendocrine carcinoma; Fig. 1B). Either fresh or cryopreserved tumor material was used to initiate xenografts, with no observed effect

of prior cryopreservation on engraftment efficiency ($p = 0.69$, Chi-square test; Supplementary Fig. 2A). Quality control for the presence of human lymphocytic tumors[13,14] revealed that 16 xenografts were human CD45 (hCD45)-expressing lymphoproliferations rather than keratin-expressing NSCLCs (described in detail in a previous manuscript[15]; Fig. 1B; Supplementary Fig. 3A). In all subsequent analyses, lymphoproliferations were considered as unsuccessful engraftments. One case (CRUK0885 Region 3; R3) lacked expression of either keratin or hCD45 but was deemed to be a NSCLC PDX model as the immunophenotype and tumor morphology was consistent with the diagnosed primary tumor subtype of carcinosarcoma (Supplementary Fig. 3A). hCD45-expressing cells were absent from first generation NSCLC PDX models in all cases except CRUK0816 R2, where CD45+ cells were present in the initial xenograft and declined over passages. Immunohistochemical analyses showed that these cells were CD3+ T lymphocytes (Supplementary Fig. 3B). Thus, our cohort consisted of 48 NSCLC PDX models from 22 patients with a successful engraftment rate of 50.0% at the patient level and 33.1% at the region level (Fig. 1B). Downsampling to one engraftment attempt per patient suggested that single region tumor sampling would have resulted in the generation of PDX models for a median of 14 patients (Fig. 1C) and that multi-region sampling increased the engraftment rate across all histological subtypes (Supplementary Fig. 4). Multiple, spatially distinct NSCLC PDX models were established for nine patients (median = 4 regional PDX models per patient with multiple PDX models; Fig. 1B). Mice with no apparent xenograft were terminated after a median of 306 days (range 37–402 days; Supplementary Fig. 2B). Each region-specific PDX model was propagated by transfer of xenograft fragments to naïve hosts, maintaining the models independently, exclusively in vivo and generating a large biobank of cryopreserved PDX tissue. PDX models could be re-established following cryopreservation (Supplementary Fig. 2C). Initial passage zero (P0) PDX models took a median of 85.0 days before tumor harvest (range 37–440 days; Fig. 1D) and variability of engraftment time between tumor regions from the same primary tumors was evident (Supplementary Fig. 2D). In subsequent passages, PDX growth was more rapid, with a median time to harvest of 51.0 days across passages P1-P3 (median values for P1, P2 and P3 were 49.5, 55.0 and 50.0, respectively; Fig. 1D).

In a review of PDX model histopathology, we observed high consistency between initial P0 and established P3 PDX models (Supplementary Fig. 5, Supplementary Data 1). When comparing PDX models to region-specific hematoxylin and eosin (H&E) stained sections available from patient tumors, we observed concordance for the majority of models, consistent with prior PDX models that have been shown to broadly resemble the histologies of the tumors from which they were derived[6,16–24]. However, in a minority of cases, we noted histological variation. Some models showed evidence of divergence at P0; for example, CRUK0949 R1 and R3 showed more widespread clear cell differentiation than was present in the corresponding patient samples for those regions, and CRUK0816 R2 and R5 PDX models presented more epithelioid differentiation than the parent tumor (Supplementary Fig. 6A). Other models varied between the initial P0 and established P3 samples, with the initial P0 PDX model more closely resembling the patient region than the P3 PDX model; for example, CRUK0941 R2 PDX model showed prominent rhabdoid differentiation at P3 that had not been present in either the patient or P0 samples (Supplementary Fig. 6B), though this was consistent with the cytological pleomorphism seen in this poorly differentiated pleomorphic carcinoma. In multiple CRUK0606 regional PDX models, substantial variation between either tumor and P0 PDX models, or P0 and P3 PDX models was observed. Glandular features were a minor component of the patient's regional tissue but became more prominent in PDX models, either in both initial P0 and established P3 models (R5, R8) or in the established P3 model only (R1, R6; Supplementary Fig. 6B).

## Engraftment characteristics of PDX-forming NSCLC tumor regions

Consistent with previous reports suggesting that PDX establishment is linked to poor prognosis in NSCLC[18,25–27], we observed a trend towards shorter disease-free survival in patients for whom at least one PDX model was established (Log rank test, $p = 0.098$; Fig. 1E). Univariate

analysis of clinical characteristics showed that lesion size was significantly associated with PDX engraftment in LUAD ($p = 0.0072$, Wilcoxon rank sum test) and other NSCLC histologies ($p = 0.024$, Wilcoxon rank sum test), but not LUSC tumors ($p = 0.95$, Wilcoxon rank sum test; Supplementary Fig. 7A). Consistent with this, an association between higher T stage (to which lesion size is a major

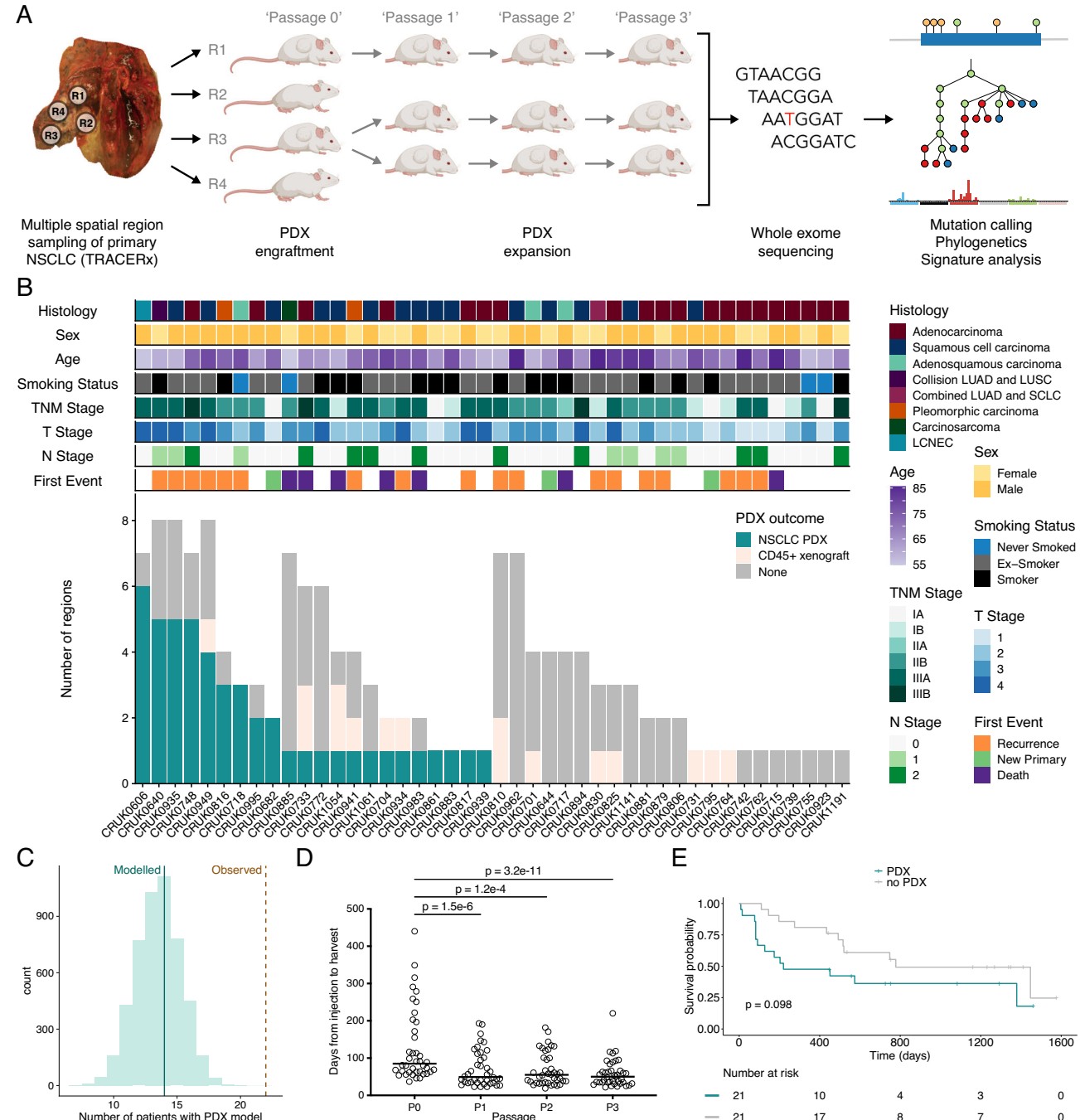

**Fig. 1 | Lung TRACERx patient-derived xenograft (PDX) cohort overview.**
**A** Schematic of the study protocol to derive and expand PDX models within the lung TRACERx study. **B** Outcomes of regional non-small cell lung cancer (NSCLC) tumor tissue engraftment in NSG mice, including patient characteristics.
**C** Downsampling to one engraftment attempt for each patient. Green line indicates median modeled number of patients with a PDX model following a single engraftment attempt, brown dashed line indicates the observed number of patients for whom PDX models were derived with a multi-region sampling approach. **D** Time from tumor injection to PDX harvest by passage number. Only PDX models for

which complete P0-P3 data were available are shown ($n = 40$ PDX models at each passage). Bar shows median time for all models. Two-sided Friedman test with Dunn's test for multiple comparisons, $p$ values as indicated. **E** Disease-free survival over a 1600 day period following tumor resection is shown grouped by the generation (PDX) or not (no PDX) of at least one regional NSCLC PDX model for each patient. Log rank test, $p$ value as indicated. LUAD−lung adenocarcinoma; LUSC− lung squamous cell carcinoma; SCLC−small cell lung cancer; LCNEC−large cell neuroendocrine carcinoma.

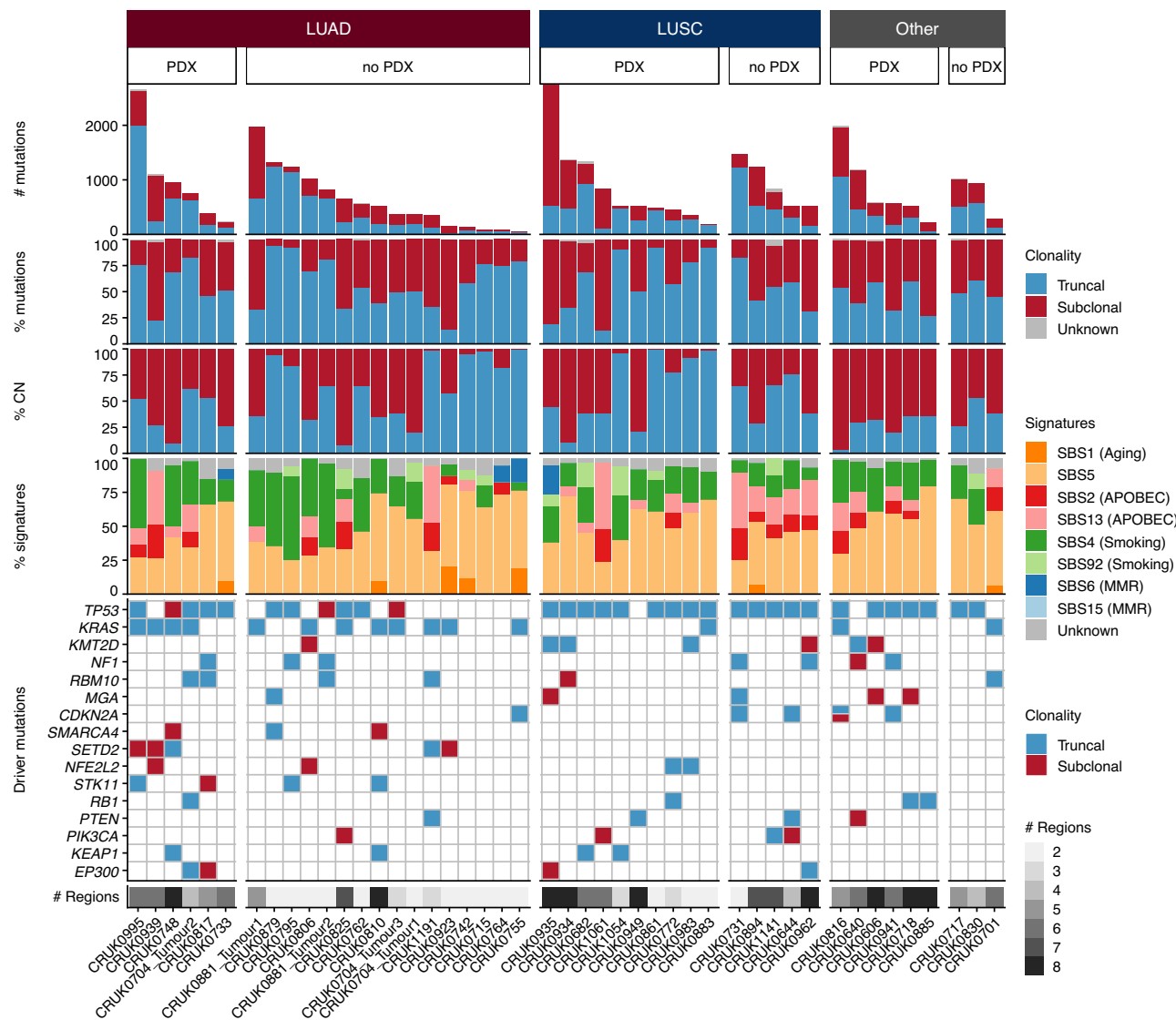

**Fig. 2 | Genomic characteristics of primary tumors.** Patient primary tumors (*n* patients = 43; *n* tumors = 46) are split based on histology and subsequently whether a PDX model was generated from any tumor region (PDX) or not (no PDX). Within each category, tumors are ordered according to their total mutation burden. Top panel: total number of coding and non-coding mutations including single nucleotide variants (SNVs), dinucleotide and indel alterations. Bars are colored by the clonality status of alterations. Second panel: proportion of truncal and subclonal mutations. Third panel: proportion of copy number alterations that were truncal or subclonal. Fourth panel: proportion of mutational signatures as estimated across all mutations. Bottom panel: driver alterations on a per tumor basis. The genes shown are mutated in more than three tumors in this patient cohort. Mutations are colored by the clonal status of alterations. LUAD—lung adenocarcinoma; LUSC—lung squamous cell carcinoma; CN—copy number; MMR—mismatch repair.

contributor) and engraftment was observed in LUAD (*p* = 0.044, Fisher's exact test; Supplementary Fig. 7E). No significant differences in overall age, smoking pack years, sex, TNM stage, pleural invasion, vascular invasion, N stage were observed between tumors that engrafted and those that did not (Supplementary Fig. 7B–I). PDX models were established for six of 20 (30.0%) LUAD patients, ten of 15 (66.7%) LUSC patients and six of nine (66.7%) patients with other NSCLC histologies (*p* = 0.064, Fisher's exact test comparing engraftment success versus histology; Supplementary Fig. 7J), consistent with literature reports of greater engraftment rates for LUSC compared to LUAD tumors (Supplementary Fig. 7K[16–18,21,23,26–32]). However, this patient-level analysis was complicated by our multiple sampling of tumors; when considering engraftment by tumor region, 11/45 LUAD (24.4%), 18/56 LUSC (32.1%) and 19/44 'other' (43.2%) regions formed PDX models (*p* = 0.18, Fisher's exact test comparing engraftment success versus histology; Supplementary Fig. 7J).

Leveraging primary tumor WES data from 43 patients (46 genomically distinct tumors), we investigated potential genomic differences between tumors that engrafted and those that did not. We found no differences in the overall number of mutations or the number of truncal and subclonal mutations (Fig. 2; Supplementary Fig. 8A-8B). Overall, differences in mutational signatures were minimal (Fig. 2), although we observed a higher number of mutations associated with APOBEC mutagenesis (SBS2 and SBS13) in LUSC tumors that did not engraft (*p* = 0.011, Wilcoxon rank sum test) and a trend was observed in the opposite direction in LUAD tumors (*p* = 0.12, Wilcoxon rank sum test; Supplementary Fig. 8C). Clustering of all samples based on total copy number led to grouping primarily by whole-genome doubling status and then by tumor of origin (Supplementary Fig. 9A, B). Regardless of engraftment success, most LUAD tumor regions analyzed were subject to whole-genome doubling. Approximately half of LUSC regions were genome-doubled, again regardless of

engraftment success. However, among regions from other NSCLC histologies, we noted a significantly higher proportion of genome-doubled regions among those that did not engraft ($p = 5.24e{-}6$, Fisher's exact test; Supplementary Fig. 9C).

We found that LUAD tumor regions that engrafted had higher copy number instability than those that did not. Specifically, the proportion of the genome that was aberrant in LUAD tumors that engrafted was significantly higher than in those that did not ($p = 0.010$ and $p = 0.00096$, tumor level and region level, respectively, Wilcoxon rank sum test; Supplementary Fig. 10A). The same pattern was observed when only considering subclonal copy number alterations. However, when considering the proportion of the aberrant genome that was subclonal, a metric that was previously associated with disease-free survival in NSCLC[33], no differences were observed (Fig. 2; Supplementary Fig. 10B). Additionally, we found that LUAD tumor regions that gave rise to PDX models had a higher fraction of the genome subject to loss of heterozygosity (LOH) than those that did not engraft ($p = 0.0015$, Wilcoxon rank sum test; Supplementary Fig. 10C). Consistent with this and the established association between *TP53* mutations and chromosomal instability in cancer[34], *TP53* mutations were enriched in tumors that gave rise to PDX models ($p = 0.026$, Fisher's exact test; Fig. 2). Considering tumor histology revealed that this was likely driven by a decreased likelihood of PDX engraftment for *TP53* wildtype LUAD tumors ($p = 0.15$, Fisher's exact test; Supplementary Fig. 10D). Indeed, the LUAD tumors with low proportions of the genome that are aberrant (Supplementary Fig. 10E) and low proportions of LOH (Supplementary Fig. 10F) were *TP53* wildtype and did not engraft.

Considering recurrent gain and/or loss events, we found several significant differences between LUAD tumor regions that successfully engrafted and those that did not. In particular, we found losses of 1p, 8p, 12p, 15q, 18q and 20p (Supplementary Fig. 11A; $q = 0.044$ and 0.03 (1p); $q = 0.044$ (8p); $q = 0.044$ (12p); $q = 0.03$ (15q); $q = 0.044$ (18q); $q = 0.044$ (20p), Fisher's exact test with false discovery rate (FDR) correction). Additionally, we observed a focal loss event on chromosome 9 that was found in all of the LUAD tumor regions that formed PDX models and approximately half of those that did not (Supplementary Fig. 11A). Higher resolution analysis of this genomic region showed that the loss event occurs within 9p21.3-9p21.1, which contains genes including *CDKN2A*, *CDKN2B* and a cluster of type I interferon genes (Supplementary Fig. 11B). A subset of ten genes, including *CDKN2A*, *CDKN2B*, *IFNA1* and *IFNE*, were found within the same segment in all engrafting LUAD tumor regions and were significantly more likely to be lost homozygously in LUAD regions that formed PDX models compared to those that did not ($p = 0.00022$, Fisher's exact test).

As estimated from patient WES data, the tumor purity of engrafted regions was higher than for non-engrafted regions overall ($p = 0.00029$, Wilcoxon rank sum test) and in both LUAD ($p = 0.011$, Wilcoxon rank sum test) and other NSCLC histologies ($p = 0.020$, Wilcoxon rank sum test; Supplementary Fig. 12A). T cell infiltration of the primary tumor regions was lower for engrafted regions as estimated using the T cell ExTRECT tool[35] (overall $p = 0.028$, Wilcoxon rank sum test), and this was driven by LUSC tumors where T cell abundance was associated with failure for PDX models to engraft ($p = 0.015$, Wilcoxon rank sum test; Supplementary Fig. 12B). Increased T cell content might reflect a higher stroma:tumor ratio within a region or may directly reduce the viability of tumor cells. Consistent with the latter, analysis of RNA sequencing data (from the subset of regions where data were available[36]) found enrichment for apoptosis-related pathways (apoptosis Hallmark pathway[37], FDR $q = 5 \times 10^{-4}$) in regions that fail to engraft and for proliferation-related pathways in regions that engrafted (E2F targets, MYC targets v1 and G2M checkpoint Hallmark pathways[37], FDR $q = 1.1 \times 10^{-7}$, $2 \times 10^{-6}$ and $2.1 \times 10^{-5}$, respectively; Supplementary Fig. 12C).

## An NSG-adapted reference genome improves removal of contaminating mouse WES reads

We subjected PDX models to WES (median depth after mouse read removal = 397x, interquartile range = 357-448x) in order to compare their genomic features to matched primary tumor regions for which WES was available within the TRACERx study (median depth = 404x, interquartile range = 373-434x). PDX models were analyzed once upon their first engraftment in mice (P0) and again once established (P3). Quality control identified a subset of contaminating reads mapping to the mouse reference genome in PDX model sequencing data (Fig. 3A). Filtering was performed to remove these reads using the bamcmp tool[38,39] and, initially, the mouse GRCm38 (mm10; C57BL/6J strain) reference genome, which is routinely used in xenograft studies. We examined non-driver mutations that were shared across tumors (i.e. recurrent passenger alterations), reasoning that these should be infrequent. However, we found 222 instances of 38 PDX-unique, non-driver mutations in PDX models recurrent in two or more tumors (Fig. 3B). As a reference genome based on the NSG mouse strain was not available, we performed whole-genome sequencing on an NSG mouse to inform the development of an NSG-adapted reference genome. This identified 7,333,533 NSG-specific single nucleotide polymorphisms (SNPs) that were not found in the mm10 reference, 90.05% of which were homozygous (Fig. 3C). Incorporating these improved the accuracy of mouse read removal, removing 168/222 (75.7%) instances of 21 mutations, with preferential removal of those that were shared across many PDX models (median removed = 6, range removed = 2–18 vs median remaining = 2, range remaining = 2–8; Fig. 3B). The remaining 54 instances occurred across 17 shared mutations and were found in tumors with a higher degree of mouse DNA contamination in P0 PDX models (Supplementary Fig. 13A). This suggests that, while our approach successfully removes most host-reads, higher mouse contamination decreases the probability of filtering such variants without a full NSG reference assembly. To validate these findings, we applied an independent analysis workflow to seven NSCLC xenografts that were established and subjected to WES at a different center. The use of the NSG-adapted reference removed a majority of PDX-unique, non-driver mutations, many of which were overlapping between the two experiments (Supplementary Fig. 13B).

## Genomic bottlenecks on engraftment result in PDX models that are monoclonal with respect to their tumor region of origin

As an initial investigation of the suitability of our sampling approach for studying genomic representation of primary tumors in PDX models, we sequenced two samples from each of five PDX models (four P0, one P3). Where a PDX model was sampled multiple times by WES, we named these samples 'A1' and 'A2'. Intra-PDX model heterogeneity was low (Supplementary Fig. 14) so we proceeded with characterization of one sample per PDX model, a priori designating the 'A1' sample as the canonical sample for subsequent analysis.

In the knowledge that primary tumor regions are heterogeneous (i.e. consist of multiple genomic subclones), we inferred the subclonal composition of P0 PDX models relative to their primary tumor region of origin. If multiple primary tumor subclones were found in the PDX model, we defined this as polyclonal engraftment, whereas if only a single primary tumor subclone was found in the PDX model, we defined this as monoclonal engraftment (Fig. 4A). Of 42 unique P0 PDX models where WES data were available, 28 were monoclonal engraftments and 14 were polyclonal (Fig. 4B). Where the clonality of the matched primary tumor region of origin could be inferred, three monoclonal PDX models arose from homogenous primary tumor regions (i.e. the PDX model was necessarily monoclonal due to the region of origin consisting of only a single clone). Of the 33 heterogeneous primary tumor regions, 13 gave rise to polyclonal PDX models with respect to the primary tumor, meaning that 20 PDX models from heterogeneous tumor regions contained only one primary tumor

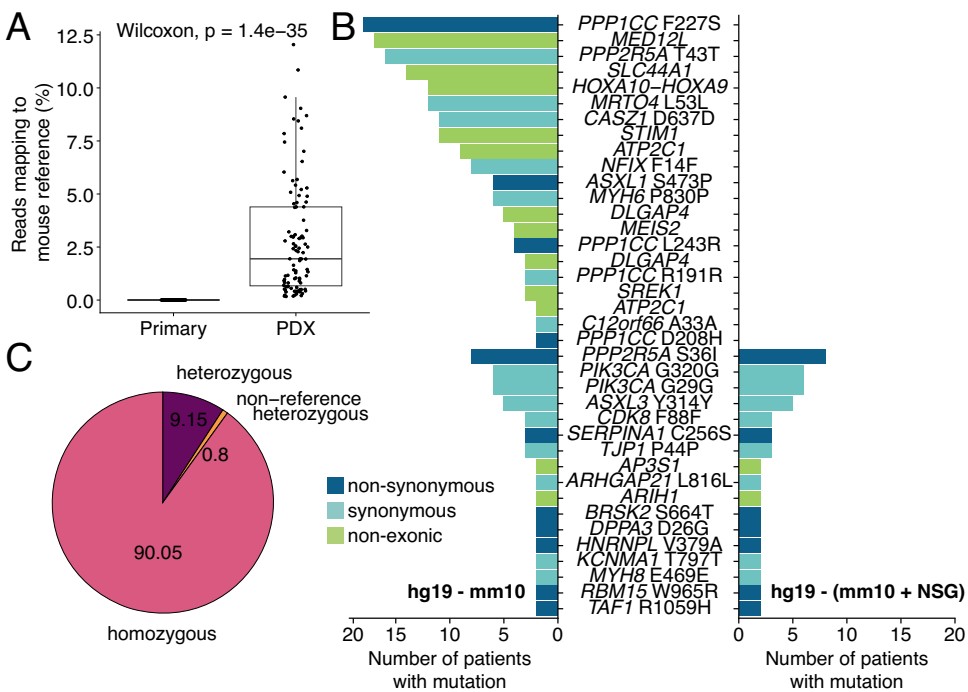

**Fig. 3 | An NSG-adapted mouse reference genome. A** Detection of mouse DNA contamination in PDX model whole-exome sequencing (WES) data using FastQ Screen (*n* primary regions = 108, *n* PDX samples = 99). The box plot represents the upper and lower quartiles (box limits), the median (center line) and the whiskers span 1.5*IQR. Two-sided Wilcoxon rank sum test, *p* value as indicated. **B** Overview of PDX-unique non-driver mutations called in PDX models from two or more patients after using the mm10 reference genome (GRCm38; left) or a NOD *scid* gamma mouse-adapted (NSG-adapted) reference genome (right) for mouse WES read removal. **C** Percentage of genome-wide single nucleotide polymorphisms (total = 7,333,533) identified as non-concordant in whole-genome sequencing data from an NSG mouse, compared to the mm10 (GRCm38) reference genome.

subclone (Supplementary Fig. 15A). These findings suggest that PDX engraftment commonly produced a bottleneck event, such that single regional PDX models frequently did not capture the genomic complexity of the primary tumor.

To further explore the similarity between the PDX models and their matched primary tumors, we calculated a mutational distance score (see Methods). This measure captures both the number of shared mutations and their prevalence for each PDX model compared to its region of origin and to all other spatially distinct regions from the same tumor. We also calculated the mutational distance between regions within each primary tumor in TRACERx421 data[33] to give an indication of diversity within primary NSCLC tumors. PDX models were significantly more similar to their region of origin than to other tumor regions from the same tumor (Fig. 4C; *p* = 8.2e−7, Wilcoxon rank sum test). Indeed, the mutational distances between PDX models and regions of the tumor other than the one used to derive the PDX model ('non-regions of origin') were comparable to those between primary tumor regions (Fig. 4C). No differences between the mutational distance of the region of origin and P0 were observed across histological subtypes (Supplementary Fig. 15B). However, we observed notable variability in the extent of similarity to the region of origin in different cases. At one extreme, the CRUK0606 R2 P0 PDX model was highly similar to its region of origin, with the lowest mutational distance within the cohort (mutation distance = 0.103). Of all mutations across the CRUK0606 R2 P0-region of origin pair, the majority of mutations were shared (68 of 111 non-truncal mutations present across the pair), with only a small number of mutations differing between the two (43/ 111; Fig. 4D, upper panel). Conversely, in CRUK0995 R3, which had the highest mutation distance between the P0 PDX and the region of origin (mutation distance = 0.819), shared mutations (83/791) were low frequency within the primary tumor and both the primary tumor region and P0 PDX model contained many additional mutations that were not shared (a total of 708/791; Fig. 4D, lower panel).

Using a comparable approach to the above for mutational distance, we calculated a copy number distance score (see Methods). We observed a significant correlation between the two metrics (Supplementary Fig. 15C; *R* = 0.71, *p* = 1.1e−6, Pearson's correlation). Similar to mutational distance, the copy number of P0 PDX models resembled their region of origin more closely than non-regions of origin (Fig. 4E; *p* = 8.3e−6, Wilcoxon rank sum test). No differences between the copy number distance of the region of origin and P0 were observed across histological subtypes (Supplementary Fig. 15D). Consistent with this, analysis of the proportion of the genome that had identical copy number between P0 PDX models and other sample types showed that P0 PDX models were frequently most similar to their matched region of origin and were more similar to non-regions of origin from their parent tumor than to regional samples from other tumors which generated at least one PDX model (Supplementary Fig. 15E). Although mutational and copy number distances followed similar patterns overall, z-transformation of the distances to compare their relative ordering across the cohort revealed that the individual PDX models with the highest mutational distances to their regions of origin did not necessarily have the highest copy number distances and vice versa (Supplementary Fig. 15F). This suggests that in specific cases, clonal selection on PDX engraftment can have different impacts on the mutational and copy number similarity to the tumor region of origin.

Next, we linked the mutational and copy number distances to PDX engraftment clonality with the hypothesis that polyclonal engraftment would better represent the heterogeneity of the tumor region of origin and therefore these metrics would be lower in those models with polyclonal P0 engraftment. Consistent with this, we found that PDX models with polyclonal engraftment trended towards a lower copy number distance to their respective region of origin than models with monoclonal engraftment from heterogeneous tumor regions (*p* = 0.057, Wilcoxon rank sum test). We did not see a difference when

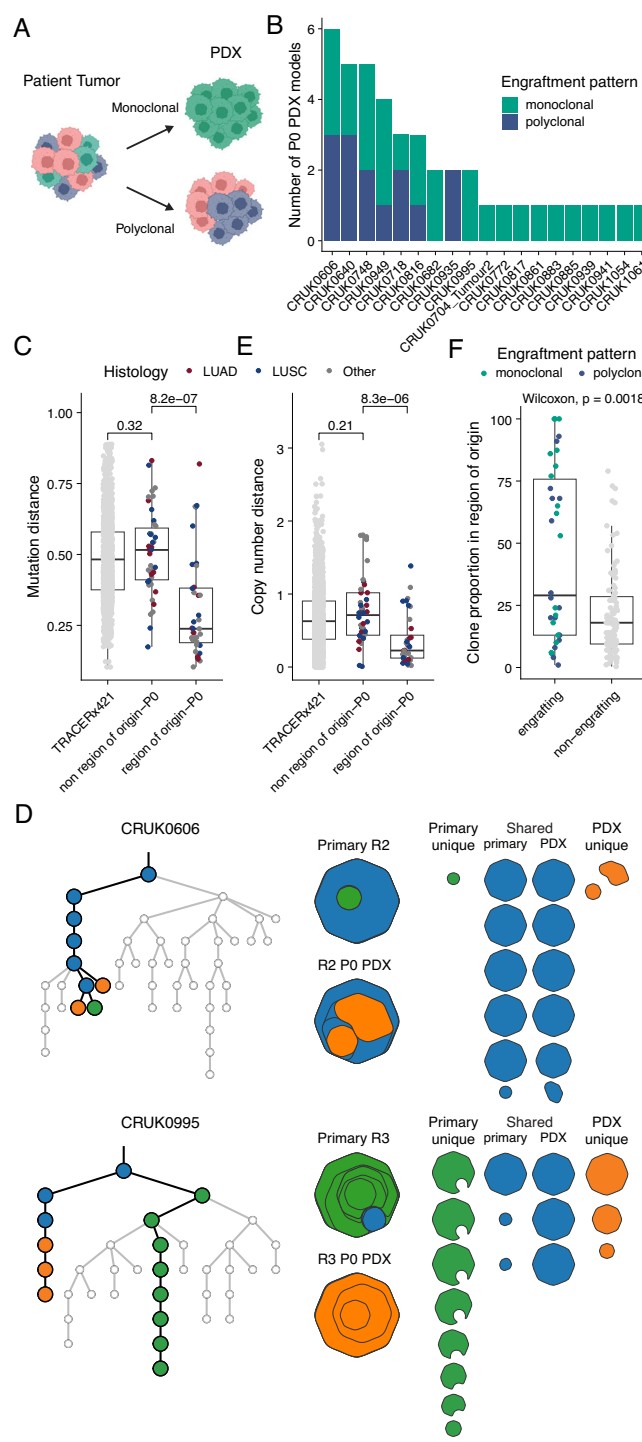

**Fig. 4 | Genomic comparison of primary tumor region-early passage PDX model pairs. A** Schematic representation of engraftment patterns. **B** Overview of engraftment patterns (monoclonal, green; polyclonal, blue) for each unique P0 PDX sample (for which WES data were available) stacked by tumor. **C** Mutational distance between regions within each primary tumor in the lung TRACERx421 cohort (*n* = 1426 regions), P0 PDX models and other regions of their primary tumor (*n* = 42 regions), and P0 PDX models and their respective region of origin (*n* = 36 regions). **D** Examples of phylogenetic trees (left) and clonal composition (right) of P0 PDX models and their regions of origin. Highlighted branches of the phylogenetic trees are those found only in the PDX models (orange), only in the region of origin (primary; green) or both (shared; blue). Clonal composition is shown as clone maps (left: stacked, right: individual clones) where each octagon corresponds to a clone from the phylogenetic tree that is present within the sample. The regions with lowest (upper: CRUK0606 R2) and greatest (lower: CRUK0995 R3) mutational distances between the region of origin and matched PDX model are shown. **E** Copy number distance between regions within each primary tumor in the lung TRA-CERx421 cohort (*n* = 1424 regions), P0 PDX models and other regions of their primary tumor (*n* = 42 regions), and P0 PDX models and their respective region of origin (*n* = 36 regions). **F** Clone proportions of engrafting (*n* = 34) and non-engrafting (*n* = 94) clones, excluding ancestral clones that were not detected, within 36 primary tumor regions of origin for which data were available. Dots are colored by the overall engraftment pattern of the P0 PDX models. **C**, **E**, **F** The box plots represent the upper and lower quartiles (box limits), the median (center line) and the whiskers span 1.5*IQR. Two-sided Wilcoxon rank sum test, *p* values as indicated. LUAD−lung adenocarcinoma; LUSC−lung squamous cell carcinoma.

considering mutational distance (*p* = 0.22, Wilcoxon rank sum test; Supplementary Fig. 15G).

To assess whether the cell populations that form a PDX model (i.e. the engrafting clone(s)) are dominant within the primary tumor region of origin, we compared the clone proportions of engrafting and non-engrafting clones within each tumor region of origin. Engrafted clones had significantly higher clone proportions compared to non-engrafting clones (*p* = 0.0018, Wilcoxon rank sum test; Fig. 4F). Despite this, we observed a bimodal distribution of engrafting clone proportions that was not driven by PDX engrafting clonality, demonstrating that minor tumor clones can contribute to monoclonal PDX model engraftment (Fig. 4F).

Although the majority of tumor region-specific PDX models were themselves heterogeneous (40/42; 95.2%), due to the dominant

monoclonal engraftment patterns observed they often represented only a single branch of the overall tumor phylogeny. In 8 of 9 cases in which we could compare multiple PDX models from the same tumor, we observed engraftment of more than one tumor subclone in independent PDX models (Fig. 5). This raised the possibility that independent PDX models from the same patient might represent different features of the primary tumor. Investigation of the distribution of driver mutations across multi-region PDX models revealed cases in which driver mutations that were subclonal in the primary tumor became fixed clonally in PDX models (Supplementary Fig. 16A). For example, CRUK0606 R3 had a subclonal *MGA* (p.G2017X) mutation in the primary tumor region, which became clonally represented in the P0 PDX model. Of note, this mutation was absent from the patient-matched R6 PDX model. Similarly, CRUK0816 R2 and R5 P0 PDX models both had a clonal *CDKN2A* (p.R46W) mutation that was absent from the R3 model, demonstrating that genomic bottlenecks during PDX engraftment can lead to altered driver mutation representation. In a third case, when considering only the primary tumor sequencing data, CRUK0995 had a truncal driver mutation in the tumor suppressor gene *STK11* (p.P179L). However, with the additional resolution afforded by sequencing PDX models, mutations with an illusion of clonality in the primary tumor could now be classified as subclonal. The *STK11* mutation was detected clonally in the R1 PDX model, consistent with the primary tumor samples, but was absent from the R3 PDX model (Supplementary Fig. 16B). This suggested that a very small ancestral subclone of the primary tumor, which had not been detected in the primary tumor sequencing data, engrafted to form the PDX model. In all three cases, these driver mutation statuses persisted in P3 models, where data were available. Thus, genomic characterization of PDX models can inform more accurate phylogenetic reconstruction of primary tumors, and monoclonal engraftment in xenografts from different tumor regions can result in PDX models with distinct driver mutation profiles.

Matched primary lymph node or recurrence/progression WES data were available for eight patients from our cohort. In an independent analysis of this patient subset, we investigated the similarity of PDX-engrafting clones and metastasis-seeding clones. Where the metastatic seeding clone(s) were present in the primary tumor at sampling (7/8 patients), metastasis seeding clones were found in at least one PDX model in 3/7 patients. In CRUK0640, the metastatic

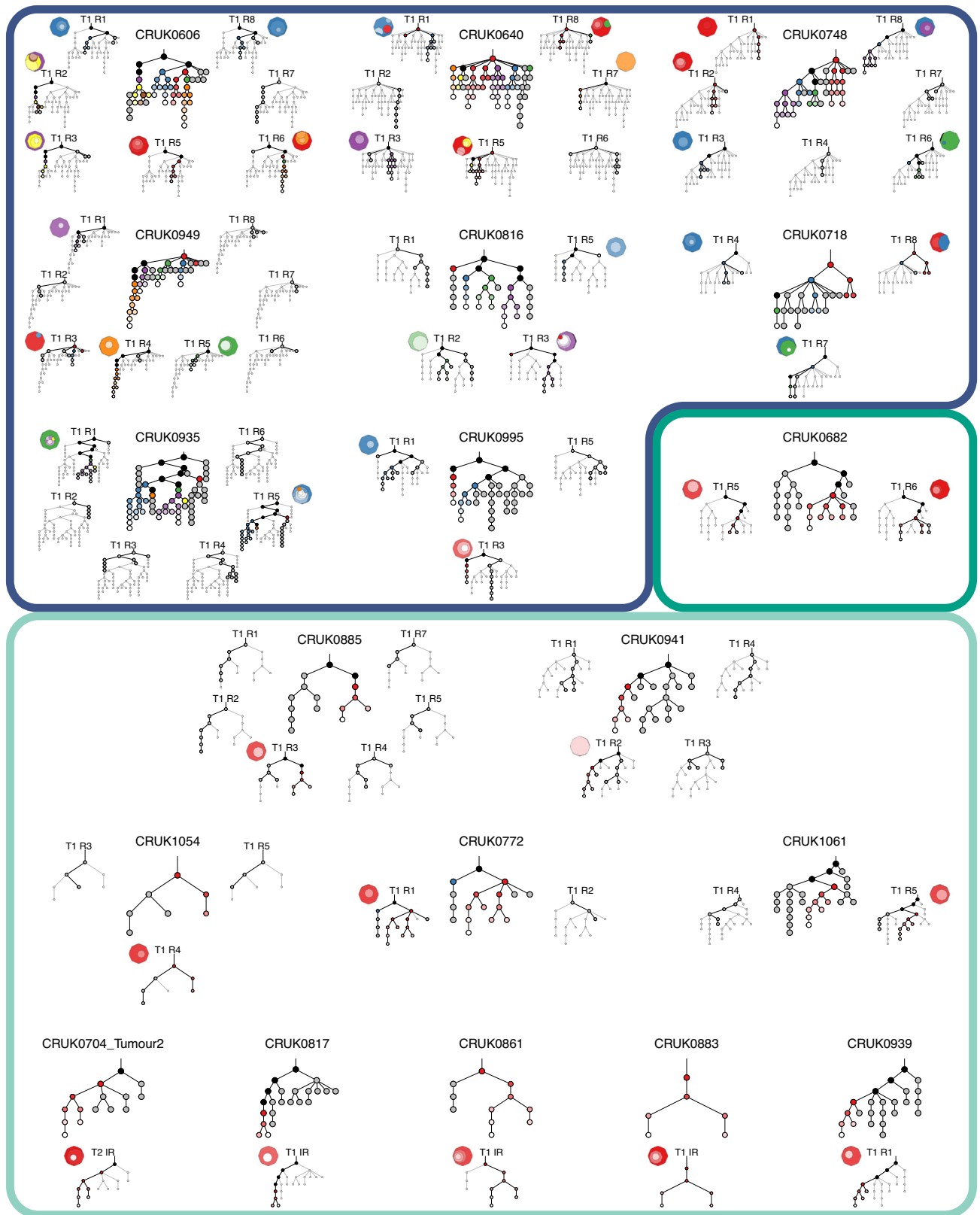

clone was found to engraft a monoclonal R7 P0 PDX model, as well as contributing to the engraftment of polyclonal P0 PDX models derived from R1, R5, and R8 but this clone did not engraft the R3 P0 PDX model. CRUK0718 also had a single metastatic clone which was found to engraft the monoclonal P0 PDX model derived from R4, and was also found to contribute to engraftment of the polyclonal R7 and R8 P0 PDX models. CRUK0748 had multiple metastatic clones of which one

was found to contribute to the engraftment of the polyclonal R6 P0 PDX model but did not engraft in the R1, R2, R3 or R8 P0 PDX models. In two further cases, the CRUK0885 R3 and CRUK0816 R2 and R5 PDX models engraftment was by the direct descendent clone of the metastasis seeding clone, although a further PDX model from CRUK0816 (R3) was engrafted by a clone on a separate branch. CRUK0941 and CRUK1061 PDX models were engrafted by clones on a

**Fig. 5 | Representation of multiple primary tumor subclones in multi-region PDX models.** An overview of phylogenetic trees (based on all primary tumor region and PDX data) and subclonal composition of the P0 PDX models is shown for each tumor that underwent whole-exome sequencing. Nine cases with multiple region-specific PDX models are shown (top). Eight of these cases have multiple primary tumor clones engrafting in the PDX models (blue border), while in one case both PDX models were engrafted by the same clone (green border). A further ten cases had a single PDX model per tumor which was engrafted by a single tumor clone (light green border, bottom). For each case, a phylogenetic tree constructed from primary tumor data and all PDX samples is shown in the center. Regional phylogenetic trees are shown for regions with attempted PDX engraftments highlighting

clusters that were present in the primary tumor region of origin and/or the matched PDX model. Black - shared clusters between primary tumor (main tree), or primary tumor region of origin (regional trees) and PDX model; gray - primary tumor specific clusters (or primary tumor region specific); colors (red, blue, green, purple, orange) indicate independent engrafting clusters, and subsequent diversification in the PDX models is indicated by a gradient of each color (to white). Clusters highlighted with a bold black border were present (either detectable as clones or ancestral) in the primary tumor (main tree) or primary tumor region of origin (regional trees) while the other clusters are either PDX-specific or below the limit of detection in the primary tumor. Additionally, for each PDX model a clone map illustrates the clonal composition of the P0 PDX sample. T−tumor; R−region.

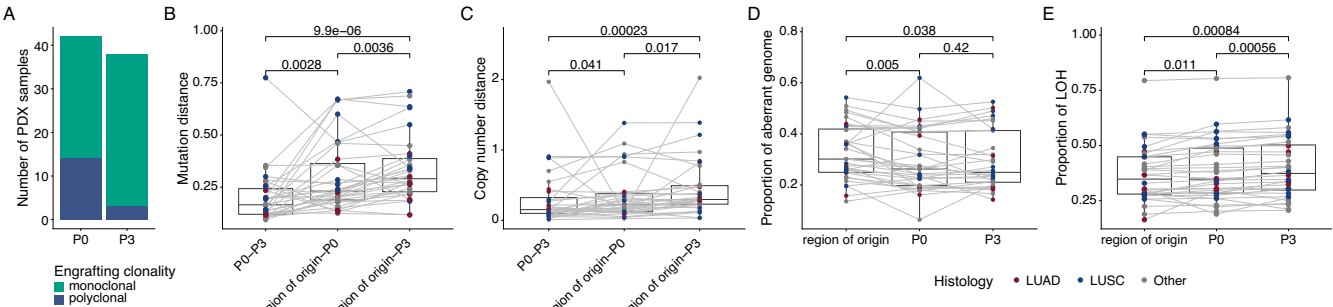

**Fig. 6 | On-going evolution in PDX models. A** Overview of engraftment patterns relative to the primary tumor of initial (P0) and established (P3) PDX models. **B**, **C** Comparison of mutational distance (**B**) and copy number distance (**C**) between P0 and matched P3 PDX models, P0 PDX models and the matched region of origin, and P3 PDX models and the matched region of origin ($n = 32$ comparisons per group). **D** The proportion of the aberrant genome for matched primary tumor region of origin, P0 PDX model and P3 PDX model samples ($n = 32$ samples per

group). **E** Proportion of the genome subject to loss of heterozygosity (LOH) for matched primary tumor region of origin, P0 PDX model and P3 PDX model samples ($n = 32$ samples per group). **B–E** The box plots represent the upper and lower quartiles (box limits), the median (center line) and the whiskers span 1.5*IQR. Lines indicate sample matching. Two-sided Wilcoxon signed rank test, $p$ values as indicated. LUAD−lung adenocarcinoma; LUSC−lung squamous cell carcinoma.

distinct branch from the metastasis. Consideration of the mutational and copy number distance scores between PDX models, primary tumor regions and metastases did not strongly support the hypothesis that PDX models from the primary tumor resemble patient metastases; while in some cases, the distances between PDX models and metastases was within the range of the distances between the PDX model and the primary tumor regions, in other cases PDX models were more dissimilar to the metastases than to any primary tumor region (Supplementary Fig. 17).

Overall, these data suggest that PDX model engraftment induces a genomic bottleneck that commonly results in engraftment of a single tumor subclone, that multiple tumor subclones can engraft in independent PDX models, and that clonal selection can alter the representation of driver mutations in PDX models. No clear signal supporting a relationship between the PDX model and metastasis seeding clones was found.

### On-going genome evolution during NSCLC PDX expansion

Our phylogenetic analyses revealed multiple clusters of PDX-unique mutations (Fig. 5), suggestive of on-going evolution distinct from the primary tumor region from which the PDX model was derived. Comparison of established P3 and matched P0 PDX models revealed that nine of 12 polyclonal P0 models (where P3 data were available) were monoclonal with respect to the primary tumor at P3 (Fig. 6A), indicating that, even if multiple subclones had initially engrafted, some PDX models had lost subclonal complexity reflective of the primary tumor region over passaging. Consistent with this, the mutational distance of P0-P3 PDX pairs was significantly lower than that from the region of origin-P0 pairs ($p = 0.0028$, Wilcoxon signed rank test), as well as region of origin-P3 pairs ($p = 9.9e−6$, Wilcoxon signed rank test; Fig. 6B), likely due to the substantial initial genetic bottleneck (Fig. 4). The same pattern was observed for copy number distance ($p = 0.041$ and $p = 0.00023$, respectively, Wilcoxon signed rank test; Fig. 6C). PDX

models where WES was performed at each passage supported the notion that initial engraftment represented a strong bottleneck in terms of both mutational (Supplementary Fig. 18A) and copy number diversity (Supplementary Fig. 18B), but these distances were more stable once PDX models were established.

In PDX models that had sufficient numbers of unique mutations, we performed mutational signature analysis to determine whether specific biological processes underlie the acquisition of new mutations in PDX models during passaging. In CRUK0935, primary tumor regions were mismatch repair (MMR) deficient and analysis of the PDX-unique mutations found in P0 and P3 models for both R1 and R5 showed evidence of on-going acquisition of mutations linked to MMR deficiency (Supplementary Fig. 18C). CRUK0995 R1 had evidence of APOBEC signature mutations (SBS2 and SBS13) in both the primary tumor-unique and the matched P3 PDX model-unique mutations, indicative of APOBEC-induced mutagenesis in the primary tumor and during PDX expansion (Supplementary Fig. 18C). In CRUK0748 R1 and R6 P3 PDX models we observed a large number of P3-unique mutations that were related to clock-like signatures (Supplementary Fig. 18C). These data suggest that tumor-intrinsic mutational signatures are active in at least some PDX models.

Comparison of the representation of driver mutations between P0 and P3 PDX models revealed that all driver mutations that were subclonal in the primary tumor and present in P0 PDX models persisted in P3 PDX models (16/16), with two of these (a *KMT2D* mutation in CRUK0606 R6 and a *NOTCH1* mutation in CRUK0816 R3) becoming clonally represented in P3 PDX models (Supplementary Fig. 18D). Additionally, we noted the emergence of five driver mutations (affecting *TP53*, *NOTCH1*, *PTEN* and *NF1*) in P3 PDX models that had been detected in neither the primary tumor nor the P0 PDX model (Supplementary Fig. 18D).

Analysis of the proportion of the genome that was aberrant revealed significant differences between the region of origin and P0 PDX models ($p = 0.0050$, Wilcoxon signed rank test), as well as

between the region of origin and P3 PDX models ($p = 0.038$, Wilcoxon signed rank test) with no differences between P0 and P3 PDX models ($p = 0.42$, Wilcoxon signed rank test; Fig. 6D). This effect was seen most strongly in 'other' NSCLC histologies (Supplementary Fig. 19A) and supports our finding that PDX models are often less heterogeneous than their tumor region of origin. Next, we considered loss of heterozygosity (LOH), which is an irreversible event: once a cell undergoes LOH, the affected genomic region cannot be reacquired. We observed slight but significant increases in the proportion of the genome affected by LOH from the region of origin to P0 ($p = 0.011$, Wilcoxon signed rank test) and from P0 to P3 PDX models ($p = 0.00056$, Wilcoxon signed rank test; Fig. 6E), with the clearest effect seen in LUSC models (Supplementary Fig. 19B). Conversely, we also investigated the phenomenon of LOH reversion, in which LOH events that are called in an early sample are not called in a related downstream sample[9]. Given the irreversibility of LOH in individual cells, this can only be explained by the expansion of a minor cell population that did not undergo LOH. The overall proportion of LOH events subject to apparent reversion was low and occurred predominantly in the region of origin-P0 comparisons in LUSC and other histologies (Supplementary Fig. 19C). In one outlier case, CRUK1054 R4, 21.7% of LOH events from the primary region were absent from the P0 PDX model. In this case, the engrafting clone was not detected in the primary tumor region suggesting engraftment of a minor ancestral clone.

To further investigate on-going copy number heterogeneity, we determined the proportion of allelic imbalance that was subject to mirrored subclonal allelic imbalance (MSAI)[40] in region of origin-P0 and P0-P3 comparisons. There was significantly more MSAI in the region of origin-P0 than P0-P3 comparisons, although some MSAI events were also observed over early PDX passages (Supplementary Fig. 19D). In one outlier model, CRUK0748 R1, 14.3% of allelic imbalance was affected by MSAI between P0 and P3 (Supplementary Fig. 19E), suggesting a high rate of on-going copy number evolution. Considering the proportion of the genome with identical copy number states, we find that P0 models are generally more similar than P3 models to the region of origin, although overall most P0 and P3 model pairs are consistent with each other (Supplementary Fig. 19F). Notably, the outliers include CRUK0748 R1, which we identified as having high levels of MSAI, and CRUK0606 R6 where we inferred a whole-genome doubling event between P0 and P3 (Supplementary Fig. 19F).

Taken together, these data suggest that, although we find on-going evolution of NSCLC PDX models over mouse passaging, the extent to which this contributes to differences between PDX models and their tumor region of origin is generally small in comparison to those caused by the genomic bottleneck on initial engraftment.

## Discussion

Here we investigated the genomic evolution of NSCLC during subcutaneous engraftment and propagation in immunodeficient mice. Previous studies based on gene expression profiling, SNP array, panel sequencing and/or whole-genome sequencing have demonstrated widespread conservation of the genomic landscape in PDX models from a range of cancer types[6,8,41]. There have been conflicting reports about the extent of on-going genomic evolution within PDX models, with different analyses suggesting that genetic drift in PDX models is either minimal[3,11,42] or substantial[9,10]. To address these issues, we developed a PDX collection within the context of a NSCLC patient cohort for whom detailed annotation, including multi-region primary tumor WES, was available for comparison. Our findings implicate a genomic bottleneck upon PDX engraftment as the major source of genomic variability between PDX models and their associated tumor region. Heterogeneous primary tumor regions often generated PDX models that were monoclonal with respect to the primary tumor, and distinct tumor clones gave rise to PDX models from different spatial regions of the primary tumor.

Quality control to ensure model and data validity are key components of PDX model pipelines. Our findings regarding the formation of B lymphoproliferations[15] are mirrored in previous PDX studies in NSCLC[13] and other cancer types[43]. These are thought to arise from EBV-transformed B cells within transplanted material whose expansion is prevented by host immune surveillance but enabled following transplantation in immunodeficient mice[44]. Measures to ensure authentic engraftment of the tissue of interest in xenograft studies are therefore essential, and, since murine lymphomas can also be transferred during subsequent passaging[45], regular surveillance for CD45+ xenografts is required. For sequencing data analysis, PDX workflows typically include a step to remove contaminating mouse reads (e.g. using bamcmp[38], Xenome[46], or other tools[47-49]). We identify mutation calls that arise in PDX samples as a result of the different SNP profiles of contaminating NSG mouse DNA and the mm10 reference genome used to identify contaminating WES reads, which is based on the C57Bl/6 J strain. By adapting the mm10 reference genome and spiking in the divergent NSG SNPs, we generated an improved filtering method. Despite this progress, our data support the need for the derivation of a complete NSG reference genome assembly for use in xenograft studies.

Few previous studies have established multiple PDX models per primary tumor, particularly in the context of matched patient tumor sequencing data. One common conclusion of studies based on single-region PDX models has been that engraftment success or failure can represent the behavior of the tumor overall. However, we find that distinct spatial regions of the same tumor can have divergent outcomes in PDX models. For example, prior studies suggest that lung squamous carcinomas more readily give rise to PDX than lung adenocarcinomas[16-18,21,23,26-32]. Our patient-level engraftment rates were consistent with this, but the LUSC engraftment rate was substantially lower at the region level than the patient level. This suggests that PDX engraftment potential might be more spatially variable in LUSC tumors, although other sampling biases (e.g. higher tissue availability from larger tumors) might play a role in the apparent histology-dependent changes in engraftment rates seen in other studies and a relatively small number of LUSC tumors (n = 15 patients, 56 regions) were analyzed here. In LUAD, successful engraftment correlated with larger tumor size (and relatedly higher T stage), a higher proportion of the genome with aberrant copy number, the presence of TP53 mutations and homozygous loss of a genomic region containing CDKN2A, CDKN2B and a cluster of type I interferon genes. TP53 mutations have been associated with better engraftment of EGFR-mutant lung adenocarcinomas in PDX models previously[50] and we speculate that, in LUAD tumors, higher chromosomal instability might represent an advantage in adapting to novel environments. In LUSC, T cell infiltration was negatively correlated with PDX engraftment, consistent with a study of breast cancer PDX models[51].

Previous studies using WES or whole-genome sequencing have typically found the conservation of a majority of tumor mutations in PDX models, but are often limited in their ability to call subclonal mutations by a low depth of coverage and a lack of comprehensive patient tumor sampling. Here, by using the same sequencing methods as were used in the TRACERx study, we could confidently call subclonal mutations. In the TRACERx 421 study, we identified a median of 8 (interquartile range, IQR: 5-12) subclones per tumor and a median of 3 (IQR: 2-4) subclones per tumor region, but the majority of PDX models in the current study were monoclonal with respect to the primary tumor. Even those that were polyclonal at P0 consisted of a median of 2 (IQR: 1.25-2) subclones that were present in the primary tumor. Therefore, a single NSCLC PDX model only modestly represents the subclonal diversity of a primary tumor. In a small number of cases, these genomic bottleneck events resulted in the derivation of PDX models with a different complement of driver alterations from the same tumor (seen in this study affecting CDKN2A, STK11 and MGA).

This may provide an explanation for the ~10% discordance in driver mutations seen between patient-PDX pairs in a recent pan-cancer study[52]. It is possible that the presence or absence of these mutations affects the drug sensitivity profiles of the PDX models; for example, mutation of *STK11*/*Lkb1*, which was found in one CRUK0995 PDX model but not the other, is known to modulate therapy response[53]. Such bottlenecking represents a potential limitation of PDX models, and should also be a consideration in approaches that use PDX models to derive cell lines or organoids for further study[7,54,55]. Moreover, the monoclonality of NSCLC PDX models with respect to the primary tumor might be a relevant consideration in designing personalized therapy approaches where representation of subclonal events could affect the therapy response of models and/or the routes available for therapy resistance.

PDX models more closely represented the tumor region from which they were derived than spatially distinct tumor regions and multiple primary tumor subclones were capable of engraftment in different engraftment attempts. The development of libraries of multiple PDX models per patient might therefore improve the capture of intratumor heterogeneity. Our data also suggest that the PDX engrafting clone cannot easily predict primary tumor clones with metastatic potential in patients.

We noted that the clonal architecture of established P3 PDX models could still be complex as a result of PDX-unique mutations that were not found in the primary tumor. These mutations suggested that genomic evolution was on-going in the PDX models, and we identified models that were defined by specific mutational signatures, such as mismatch repair deficiency and APOBEC mutagenesis. Overall, the on-going accumulation of mutations over approximately eight months of expansion in mice contributed less to the overall genomic distance of PDX models from primary tumors than initial bottlenecking events. A caveat of our study is that we did not analyze PDX tumors after P3, meaning that we cannot assess the genomic evolution in later passage models, which are frequently used in the literature. Nevertheless, the on-going evolution of models suggests the value of generating banks of low passage PDX models and regular screening of PDX cohorts for acquired genomic changes.

In summary, by tracking cancer mutations through primary NSCLC engraftment and expansion in PDX models, we reveal a genomic bottleneck during engraftment that often means an individual PDX model is representative of only one subclone of the primary tumor. The full representation of truncal tumor alterations (those present in all cells of the tumor) in PDX models supports their use in cohort level studies and for testing therapeutics targeting truncal events. However, the underrepresentation of subclonal heterogeneity in PDX models suggests that care should be taken in extrapolating data from single region PDX models in experiments where subclonal events (i.e. intratumor heterogeneity) might be significant for the outcome (e.g. in personalized medicine approaches[56]). Experimental approaches to assess the functional consequences of PDX monoclonality, particularly for therapy response and resistance, are now required.

## Methods

### Generation and maintenance of multi-region NSCLC PDX models

Ethical approval to generate patient-derived models was obtained through the Tracking Cancer Evolution through Therapy (TRACERx) clinical study (REC reference: 13/LO/1546; https://clinicaltrials.gov/ct2/show/NCT01888601). Animal studies were approved by the University College London Biological Services Ethical Review Committee and licensed under UK Home Office regulations (project license P36565407).

Tissue from patients undergoing surgical resection of NSCLCs was immediately transported on ice from the operating room to a pathology laboratory where it was dissected for diagnostic and then research purposes. Tumor samples were dissected by a consultant pathologist such that the tissue used to generate patient-derived xenograft (PDX) models was considered to be within the same tumor region as material sequenced in the TRACERx study. In cases where region-matched tissue could not be collected for PDX studies, inter-region (IR) tumor tissue was used. Tumor samples for PDX studies were transported to the laboratory in a transport medium consisting of MEM alpha medium (Gibco) containing 1X penicillin/streptomycin (Gibco), 1X gentamicin (Gibco) and 1X amphotericin B (Fisher Scientific, UK). Samples were minced using a scalpel and either resuspended in 180 µl growth factor-reduced Matrigel (BD Biosciences) for fresh injection, or frozen in ice-cold fetal bovine serum plus 10% DMSO, first to −80 °C in a CoolCell (Corning) before long-term storage in liquid nitrogen.

Male non-obese diabetic/severe combined immunodeficient (NOD *scid* gamma; NSG) mice were housed in individually ventilated cages under specific pathogen-free conditions and had ad libitum access to sterile food and water. The room housing the mice was maintained on a 12 h light-dark cycle (with gradually increasing light from 6:30 am to 7:00 am and gradually decreasing light from 6:30 to 7:00 pm). Temperature was maintained in a 20–24 °C range and humidity was maintained at 55% (±10%). Mice were typically between 6 and 12 weeks of age at the time of tumor/PDX implantation. To generate PDX tumors, mice were anesthetized using 2–4% isoflurane, the flank was shaved and cleaned before tumor tissue in Matrigel was injected subcutaneously using a 16 G needle. Mice were observed during recovery, then monitored twice per week for tumor growth. When xenograft tumors formed, tumor measurements were taken in two dimensions using calipers and mice were euthanized before tumors reached 1.5 cm³ in volume (this animal license limit was not exceeded). Mice without xenograft tumors were terminated after a median of 306 days (range 37–402 days). Successfully engrafted tumors were propagated through four generations of mice, with banking of histology tissue, OCT-embedded frozen tissue and xenograft DNA at each generation. Cryopreservation of living xenograft tissue was also performed at each tumor transfer as per patient tissue.

### Histopathological characterization

Paraffin-fixed tissue sections were routinely obtained at PDX passage by fixation of tumor fragments (approximately 3 × 3 × 3 mm in size) in 4% paraformaldehyde. Samples were fixed overnight at 4 °C and stored in 70% ethanol at 4 °C before being processed through an ethanol gradient using an automated pipeline and embedded in paraffin. Formalin-fixed paraffin-embedded tissue sections of PDX tumors and their equivalent primary tumor region were subjected to hematoxylin and eosin (H&E) staining or immunohistochemistry with the following antibodies; anti-CD45 (Clone HI30; Dilution 1:200; Cat No 304002); anti-keratin (Clone: AE1/AE3; Dilution: 1:100; Cat No: 13160); anti-CD3 (Clone: LN10; Dilution: 1:100; Cat No: NCL-L-CD3-565); anti-CD20 (Clone L26; Dilution: 1:200; Cat No: M0755). Optimization of the antibodies was carried out on sections of human tonsil tissues. Immunostaining was performed using an automated BOND-III Autostainer (Leica Microsystems, UK) according to protocols described previously[57]. Slide images were acquired using a NanoZoomer 2.0HT whole slide imaging system (Hamamatsu Photonics, Japan). Figure panels containing overview images with a selected region of interest were generated semi-automatically from Nanozoomer ndpi whole-slide digital images using the PATHOverview tool (https://github.com/EpiCENTR-Lab/PATHOverview)[58].

Slides from P0 and P3 PDX models, along with region-specific H&E images from the patient tumor, were subjected to a comprehensive pathology review by a consultant pathologist. Instances in which samples were consistent with one another were scored 2 (consistent), instances in which broad similarity of histopathological subtype were observed but with minor differences, for example in the prevalence of

a particular growth pattern, were scored 1 (divergent), while samples that were dissimilar from one another were scored 0 (inconsistent). Differences in the extent of necrosis between samples were common but were ignored for scoring purposes as it is likely to be affected by experimental factors in addition to being a characteristic of specific tumor regions/PDX models.

### Genomic profiling

DNA was extracted from PDX models at each transfer using either the PureLink Genomic DNA Mini Kit (Invitrogen) or the DNA/RNA AllPrep Kit (Qiagen). For each PDX sample, exome capture was performed on 200 ng DNA using a customized version of the Agilent Human All Exome V5 Kit (Agilent) according to the manufacturer's protocol, as previously reported[40]. Following cluster generation, samples were 100 bp paired-end multiplex sequenced on the Illumina HiSeq 2500 and HiSeq 4000 at the Advanced Sequencing Facility at The Francis Crick Institute, London, U.K. Protocols for DNA extraction and processing of the tumor samples have been previously reported[33,40].

### Generation of independent PDX model WES data for validation of the NSG-adapted reference genome

At a different center, xenograft models were generated by the implantation of fresh tumor tissue from resected patient lung tumors subcutaneously into NSG mice. If xenografts formed, these were passed into subsequent mice up to three times. Either snap-frozen xenograft tissue or xenograft tissue stored in RNAlater was used to extract nucleic acids with a QIAGEN kit. DNA was also isolated from matched patient white blood cell samples for use as germline controls. WES target enrichment was performed using the Agilent SureSelectXT Human All Exon V6 Capture Library for all samples, followed by paired-end 2 × 101 bp sequencing on a NovaSeq sequencer.

### Bioinformatics pipeline

**Alignment.** Initial quality control of raw paired-end reads (100 bp) was performed using FastQC (0.11.8, https://www.bioinformatics. babraham.ac.uk/projects/fastqc/) and FastQ Screen (0.13.0, https:// www.bioinformatics.babraham.ac.uk/projects/fastq_screen/, flags: --subset 100000; --aligner bowtie2). Subsequently, fastp (0.20.0, flags: --length_required 36; --cut_window_size 4; --cut_mean_quality 10; --average_qual 20) was used to remove adapter sequences and quality trim reads. Trimmed reads were aligned to the hg19 genome assembly (including unknown contigs) using BWA-MEM (0.7.17)[59,60]. Alignments were performed separately for each lane of sequencing and then merged from the same patient region using Sambamba (0.7.0)[61] and deduplicated using Picard Tools (2.21.9, http://broadinstitute.github. io/picard/)[62]. Local realignment around indels was performed using the Genome Analysis Toolkit (GATK, 3.8.1)[63]. Further quality control following alignment was performed using a combination of Somalier (0.2.7, https://github.com/brentp/somalier), Samtools (1.9)[64], Picard Tools[62], and Conpair (0.2)[65].

For PDX samples, the steps above were repeated twice, aligning once to the hg19 genome assembly and once to the mm10 genome assembly or an NSG-adapted mouse reference (see below). Subsequently, bamcmp[38] (v2.1, using the alignment score metric) was used to identify contaminating mouse reads in our xenograft data. Only reads aligning solely to hg19 or better to hg19 compared to mm10 were included in subsequent downstream processing steps.

To obtain the median coverage per primary tumor or PDX sample, we used GATK DepthOfCoverage (GATK v4.2.0.0; with flags: --omit-depth-output-at-each-base true; --omit-interval-statistics true; --read-filter MappingQualityReadFilter; --minimum-mapping-quality 20; --stop 1000)[63]. This step was performed after alignment and, in the case of the PDX samples, after mouse read deconvolution using bamcmp.

**NSG-adapted reference genome.** Genomic DNA was extracted from the tail of one NSG mouse using the PureLink Genomic DNA Mini Kit (Invitrogen). Library preparation was performed on 500 ng DNA using an Illumina DNA Prep kit (#20018705) according to the manufacturer's protocol. Libraries were amplified through five cycles of PCR and sequencing was performed on a NovaSeq 6000 (100 bp paired-end reads) with a target sequencing depth of 30x. The raw paired-end reads were processed through the nf-core Sarek pipeline (v2.7.1)[66] using Nextflow (v21.01.4)[67]. The reference genome was manually specified as mm10 and subsequently GATK HaplotypeCaller was run to detect single nucleotide polymorphisms (SNPs) differing between mm10 and the sequenced NSG genome.

The output from HaplotypeCaller was subsequently filtered to remove heterozygous and only retain homozygous and non-reference heterozygous SNPs. These variants were spiked into the mm10 reference genome using BCFtools (v1.12)[68] 'consensus' specifying that the first allele should be used.

The raw sequencing output of the NSG mouse can be downloaded from ENA (accession number PRJEB65917) while the processed output from HaplotypeCaller, the NSG-adapted reference and the scripts to reproduce these can be found on Zenodo (https://doi.org/10.5281/zenodo.10304174)[69].

**Quantifying mouse contamination.** To estimate the proportion of reads mapping to human (GRCh38) and mouse (GRCm38) reference genomes in the primary tumor and PDX samples, FastQ Screen (0.13.0, https://www.bioinformatics.babraham.ac.uk/projects/fastq_screen/) was run individually on each lane of sequencing (flags: --subset 100000; --aligner bowtie2)[70]. Subsequently, search libraries that were not human or mouse were summarized as 'Other' and, for each sample, all lanes of sequencing were summarized using the median number of reads mapping to each group (Human, Mouse, Other, Multiple Genomes, No hits). The final output specifies the percentage of the query reads mapping to each group.

**Subsequent processing.** The downstream steps of somatic mutation calling and somatic copy number alteration detections, as well as manual quality control were performed analogously to the methods described previously[33].

In brief, SAMtools mpileup (v1.10) was used to locate non-reference positions in tumor and germline samples and the output was used by VarScan2 somatic (v2.4.4)[71] to identify tumor somatic variants. The resulting single nucleotide variant (SNV) calls were filtered for false positives using Varscan2's associated fpfilter.pl script. MuTect (1.1.7)[72] was also used to detect SNVs utilizing annotation files contained in GATK bundle 2.8. Following completion, variants called by MuTect were filtered according to the filter parameter 'PASS'.

Additional filtering was performed to minimize false positive variant calls. An SNV was considered a true positive if the variant allele frequency (VAF) was greater than 2% and the mutation was called by both VarScan2, with a somatic p-value ≤ 0.01, and MuTect. Alternatively, a frequency of 5% was required if only called in VarScan2, again with a somatic p-value ≤ 0.01. Additionally, sequencing depth in each region was required to be ≥30x and ≥10 sequence reads had to support the variant call. In contrast, the number of reads supporting the variant in the germline data had to be <5 and the VAF ≤ 1%.

The power of multi-region sequencing was leveraged to allow low variant frequency to be called with increased confidence: Where a somatic variant was not called ubiquitously across tumor regions but was called in one or more region, read information was extracted from the original alignment file using bam-readcount (v0.8.0, https://github.com/genome/bam-readcount). In such cases, VAF restrictions were reduced to VAF ≥ 1% allowing for the positive identification of low-frequency variants that would otherwise have been missed.

Indels were filtered using the same parameters as described above, with the exception of the requirement of ≥10 reads supporting the variant call, a somatic p-value of ≤0.001 and a sequencing depth of ≥50x. Occasionally, when attempting to identify indels across multiple tumor regions, discrepancies in the start position, end position or length of the indel were identified. In such cases, the longest predicted indel was reported and the maximum sequence-related values reported.

Dinucleotide variants were identified in cases where two adjacent SNVs were called. In such cases, a proportion test was performed to provide an indication as to whether the frequency of the two SNVs was significantly similar and thereby indicative of a single mutational event. In such cases, the start and stop position was corrected to represent a dinucleotide substitution and sequence-related values (e.g. coverage and variant count) were recalculated to represent the mean of the SNVs. Variants were annotated using Annovar[73] and COSMIC[74].

For somatic copy number alteration, heterozygous SNPs were identified from the germline samples using platypus (v0.8.1)[75]. LogR data was calculated using VarScan2[71] and GC-corrected using a wave-pattern GC correction method[76]. These data were processed using ASCAT (v2.3)[77] as well as Sequenza (v2.1.2)[78] for tumor purity and ploidy estimation. Manual verification of the automatically selected models was performed and samples that had insufficient purity (<10%) were excluded.

Subsequently, refphase[79] was used to infer haplotype-specific copy number alterations and to rescue low purity tumor regions, leveraging the multi-region data. CONIPHER[80] was used to cluster mutations and reconstruct phylogenetic trees for both the primary tumor regions and all PDX samples combined.

For analysis of driver alterations, lung cancer-associated genes were derived from Bailey et al.[81], Berger et al.[82], Martincorena et al.[83] or through de novo dN/dS discovery in Frankell et al.[33]. If the mutation was found to be deleterious (either a stop-gain or predicted deleterious in two out of the three computational approaches applied: Sift[84], Polyphen[85] and MutationTaster[86]), and the gene was annotated as being recessive in COSMIC (tumor suppressor), the variant was classified as a driver mutation. Also, if the gene was annotated as being dominant (oncogene) in COSMIC, and we could identify ≥3 exact matches of the specific variant in COSMIC, we classified the mutation as a driver mutation, as per the approach in Frankell et al.[33].

**Distinguishing multiple independent tumors from a single patient.** On a per patient basis, to determine whether multiple samples were genomically related, we performed a clustering step on the mutations identified in each region. Firstly, all ubiquitous mutations were determined that had a VAF greater than 1% in all regions. If more than ten such mutations existed, the regions were deemed genomically related. Conversely, if ten or fewer mutations were shared across all regions, a clustering step using the R function 'hclust' was performed on the mutation VAFs across all regions. Subsequently, the resulting clustering tree was separated into two groups to determine the regions associated with two distinct tumors. This step was repeated on the two distinct tumors, respectively to yield a maximum of four distinct tumors.

**Pipeline for NSG-adapted reference genome validation experiments**
For the seven NSCLC xenograft exomes used in validation experiments, adapter sequences and low quality reads were removed from the raw FASTQ files using Cutadapt (v.4.4[87], flags: --trim-n; --minimum-length=30; -e 0.3; -O 10; -n 2). Trimmed reads from each of the sequencer lanes were separately mapped to human genome reference GRCh38, mouse genome reference GRCm38 and the NSG-adapted mouse reference genome, all using bwa-mem (v.0.7.17). Potential mouse contamination reads were identified from the human reference

mapped alignments using bamcmp (v.2.2)[38], using the alignment score metric. Reads that were uniquely aligned or had a better alignment score to the human reference genome compared to the mouse genomes were included for subsequent downstream analyses (separately for GRCm38 and the NSG-adapted genome). Alignment from different lanes were merged using samtools (v.1.17)[68], duplicated reads were marked using Picard tools (v.3.0.0)[62] and base quality scores were recalibrated using GATK (v.4.4.0.0)[63].

Somatic variants were called from the mouse-filtered human alignments using Mutect2 from GATK (v.4.4.0.0). Common population variants from the gnomAD database (v.2.1.1) were used with Mutect2 to filter variants that were present in gnomAD with allele frequency greater than 0.01, followed by using GATK's FilterMutectCalls. Passed variants were annotated using VEP (v.106)[88] and were converted to MAF format using vcf2maf (v.1.6.18)[89].

## Analysis

**Downsampling engraftment attempts.** To estimate patient-level engraftment success if only a single tumor region was attempted, we performed a downsampling approach. For this, we used all attempted regions per patient (n patients = 44, n regions = 145) and the classification of whether the region successfully engrafted or not. We then randomly sampled one region per patient and summarized the number of successful engraftment attempts of the selection. This process was repeated 5000 times. For the histology specific downsampling, the same approach was taken as described above, except that successful engraftment was summarized within the individual histological subtypes for each iteration.

**Mutational signature deconvolution.** Mutational signatures were estimated at the tumor level using deconstructSigs (v1.9.0)[90] using all mutations. Mutation counts were normalized using the 'exome2genome' parameter and COSMIC mutational signatures (v3.2) were specified. Only SBS1, SBS2, SBS4, SBS5, SBS13, SBS92 which have previously been shown to be active in lung cancer[91], as well as SBS6 and SBS15 which are associated with DNA mismatch repair and were detected within one patient in the cohort were used to reconstruct mutational profiles.

To plot individual mutation profiles, the function 'mut.to.sigs.input' was used to create the mutation counts for each substitution type which were subsequently visualized using ggplot2[92].

**Weighted fraction of the genome subject to loss of heterozygosity.** The weighted fraction of the genome subject to loss of heterozygosity (wFLOH) was calculated as the mean of the proportions of LOH across each chromosome (excluding sex chromosomes).

**Somatic copy number alteration metrics.** Somatic copy number alterations (SCNAs) were defined using refphase[79] as any gain or loss occurring in a given sample. These events were defined as homogeneous if they were shared across all samples of a tumor and heterogeneous if only a subset of the samples had a gain or loss of a given segment. The proportion of the genome that is aberrant was defined as the length of segments affected by gains or losses divided by the total length of the genome. The proportion of the genome that was aberrant subclonally was calculated by dividing the total length of heterogeneous gains or losses by the total length of the genome. These measures were computed both at the tumor level and individual sample level. For the proportion of the genome aberrant subclonally at the sample level only heterogeneous events occurring in the sample were considered.

SCNA intratumor heterogeneity (ITH) was calculated for each tumor as the proportion of the genome harboring heterogeneous (i.e. subclonal) SCNA events divided by the proportion of the genome harboring any SCNA events.

For any two samples, the proportion of the genome which was identical was calculated as the total length of segments where both samples had the same total copy number divided by the total length of all segments.

The proportion of the genome with allelic imbalance affected by mirrored subclonal allelic imbalance (MSAI) was calculated as the total length of segments where allelic imbalance was detected in both samples, but affecting different alleles divided by the total length of segments affected by allelic imbalance.

The proportion of segments with loss of heterozygosity (LOH) affected by LOH reversion was calculated as the total length of segments where LOH was detected in the region of origin but was not detected in the P0 PDX model (LOH reversion) divided by the total length of segments affected by LOH. To define LOH reversion, only segments with more than 10 heterozygous SNPs and a minor allelic copy number of less than 0.1 were considered.

**Genome doubling detection.** Genome doubling status was estimated using ParallelGDDetect (https://github.com/amf71/ParallelGDDetect) analogously as described previously[33]. In brief, the genome doubling status of a sample was estimated using the genome-wide copy number of the major allele. If the major allele had a copy of ≥2 across at least 50% of the genome, the sample was considered to have undergone a whole genome doubling (WGD) event.

**Sample purity.** Sample purity was initially obtained from ASCAT (v2.3)[77] or Sequenza (v2.1.2)[78] and subject to manual review. Subsequently, refphase performed additional slight purity corrections based on the multi-region informed segmentation and phasing.

**Calculating the T cell fraction from whole-exome sequencing data.** To calculate the T cell fraction of the tumor samples from whole exome sequencing, the R package T cell ExTRECT[35] was used. Coverage values were extracted using the pre-defined TCRA gene segments ('tcra_seg_hg19'). Subsequently, T cell fractions were estimated using the pre-specified exon locations for the Agilent capture kit ('TCRA_exons_hg19'), the TCRA gene segments ('tcra_seg_hg19') and specifying the reference genome version as 'hg19'.

**Copy number analysis across the genome.** The copy number across the genome for samples that engrafted and those that did not, split by histology was visualized using complex heatmap (v2.15.4)[93]. For this, the total copy number, i.e. sum of integer copy numbers of allele A and B, was used and for visualization purposes, the total copy number was capped at 10. The histology and PDX vs no PDX groups were pre-specified.

For clustering of total copy number across the genome, a Euclidean distance matrix was calculated using the 'dist' function in R. Then, hierarchical clustering was performed using Ward's method[94] and the resulting dendrogram was split into three groups.

We split the genome into 5 megabase (Mb) bins and overlapped the segments obtained from refphase[79] for each sample with these bins. If, for a sample, any segment overlapping a bin was gained (or lost) the bin was defined as gained (or lost, as appropriate) for that sample. The proportion of samples with a gain (or loss) in a bin was plotted across the genome for LUAD and LUSC tumor regions that either successfully engrafted (PDX) or did not (no PDX).

We performed a power calculation (using the 'power.fisher.test' function from the R package statmod) to determine the minimum and maximum number of gains (losses) necessary across both PDX and no PDX groups together to achieve a power > 0.8. This resulted in a minimum of five samples with a gain (or loss) for both LUAD and LUSC and a maximum of 26 samples for LUAD and 41 samples for LUSC.

Segments with n gains (or losses) between these two thresholds were then tested using Fisher's exact tests separately for LUAD and LUSC. P values were adjusted using a false discovery rate (FDR) correction.

**Classifying engraftment patterns.** Within each primary tumor we identified which cancer clone(s) were involved in PDX engraftment and classified the engraftment pattern as monoclonal if only a single clone of the primary tumor was engrafted in PDX samples, or polyclonal if multiple cancer clones were involved in engraftment. Specifically, for each individual PDX sample, if all mutation clusters shared between the primary tumor and the sample were found to be clonal within the PDX, the engraftment pattern was defined as monoclonal. Conversely, if any cluster defined as subclonal within the PDX sample was also present in the primary tumor, the engraftment was classified as polyclonal.

If only a single PDX sample was considered for a patient, the tumor-level engraftment pattern matched the PDX-level engraftment pattern. If multiple PDX models were sampled and the engraftment pattern of any individual PDX sample was defined as polyclonal, the tumor-level engraftment pattern was also defined as polyclonal. Conversely, if all PDX samples followed a monoclonal engraftment pattern, all shared clusters between the primary tumor and each PDX were extracted. If all shared clusters overlapped across all PDX samples, the tumor-level engraftment pattern was classified as monoclonal, while if any PDX sample shared additional clusters with the primary tumor, the overall engraftment pattern was defined as polyclonal.

**Defining the engrafting clones.** The engrafting clone is defined as the most recent shared clone between the primary tumor and PDX model. Any cluster present in the primary tumor (defined as clonal or subclonal) and absent from the PDX sample was defined as primary specific, any cluster present solely in the PDX and absent from the primary tumor was defined as PDX specific, while all clusters present in both the primary tumor and PDX were defined as shared.

The shared clusters were mapped to the phylogenetic tree to determine the most recent shared cluster using a leaf-up approach. If the shared clusters could be mapped to a single branch of the phylogenetic tree, the clonality of the most recent shared cluster was determined in the PDX sample. If the most recent shared cluster was clonal in the PDX sample, this cluster was defined as the only engrafting cluster for the PDX sample. On the other hand, if the most recent shared cluster was subclonal within the PDX, the parent cluster was also considered. This was done iteratively until the first shared cluster which was clonal in the PDX was found. Clusters along this path were defined as engrafting if their phyloCCF value was greater than the phyloCCF of the child cluster.

If the shared clusters were mapped to multiple branches of the phylogenetic tree, each branch was considered separately in the manner described above. If a parent cluster was shared between multiple branches, CCF values of both branches were added together and the iterative approach continued until the first cluster was found to be clonal in the PDX sample.

**Calculating clone proportions.** The clone proportions of engrafting clones in the region of origin were calculated using the CONIPHER function 'compute_subclone_proportions'[80]. In short, the cancer cell fraction of the clone and all descendants, as well as the phylogenetic structure are considered and the proportion of cells belonging to the engrafting clone present in the sample is calculated as the total CCF of the clone, subtracting the CCF of all descendants. Notably, all subclone proportions will sum to 1 in each sample and will only correspond to the CCF in the case of leaf nodes on the phylogenetic tree. In this way, subclonal expansions in a tumor or PDX sample can be inferred, as well as whether a region is homogeneous (consisting only of a single clone) or heterogeneous (consisting of multiple subclones at the point of sampling).

**Mutational distance.** The mutational distance gives an approximation of mutational similarity between two regions, and also accounts for any large bottlenecks. Specifically, the distance will be large if few mutations are shared, or shared mutations occur at very different cellular frequencies; while the distance will be small if most mutations occur at similar frequencies across two regions.

Given two regions $i$ and $j$, and $M$ being the total number of mutations present in either one or the other region, excluding all truncal mutations; the mutation distance is calculated as:

$$\frac{1}{M}\sum_{m=1}^{M}|CCF_{i,m} - CCF_{j,m}|$$

Where $CCF_{i,m}$ and $CCF_{j,m}$ are the CCF of mutation $m$ in region $i$ or $j$, respectively.

To calculate a distance for each region, the pairwise distance to each other region of interest is calculated and the average across all pairwise distances is computed.

**Copy number distance.** The copy number distance gives an approximation of similarity between two regions relating to relative gains and losses of segments. If gains/losses of segments relative to ploidy are consistent across two regions the copy number distance is small; whereas when they diverge, e.g. a loss in one region and neutral copy number state in the other, the distance increases.

Given two regions $i$ and $j$, and $S$ being the total number of aberrant segments in either one or the other region, excluding all truncal copy number alterations; the copy number distance is calculated as:

$$\frac{1}{\sum_{s=1}^{S}l_s}*\sum_{s=1}^{S}l_s*|CN_{i,s} - CN_{j,s}|$$

Where $CN_{i,s}$ and $CN_{j,s}$ are the total copy number of segment $s$ in region $i$ or $j$, respectively, and $l_s$ is the length of segment $s$.

**Comparing mutational and copy number distances.** In order to classify PDX models whose bottleneck event upon engraftment was characterized by predominantly mutation versus copy number events, both distances were z-transformed. PDX models from the upper and lower quartiles of the difference between mutation and copy number distances were classified as higher mutation or copy number diversity, respectively.

**Depiction of clonal composition in tumor samples using clone maps.** In Figs. 4 and 5, the clonal composition of tumor samples are estimated using CONIPHER[80], accounting for the nesting structure determined by the phylogenetic tree building. The images were generated using the cloneMap R package (version 1.0.0) available on github (https://github.com/amf71/cloneMap).

**Differential gene expression and gene set enrichment analysis.** Differential gene expression and subsequent gene set enrichment analyses (GSEA) were performed on the samples from this study that had available RNA-seq in the TRACERx 421 cohort[36] (32 primary tumor samples from 8 patients, of which 10 samples generated a PDX model and 22 generated no PDX), using the following approach. First, trimmed mean of M-values normalization from the edgeR (v.3.32.0[95]) R package was performed on RSEM raw counts. Genes with expression below 30 counts per million in <70% of the smallest group size were removed using the function filterByExpr() with min.count set to 30. Expression differences were performed at the region level through the limma-voom analytical pipeline, using the sex of the patient as a covariate and taking patient of origin as a blocking factor, by performing within-tumor expression correlations and including them within the voom model estimate using the duplicateCorrelation() function. This method is analogous to using tumor as a random effect in a linear mixed-effects model. The raw $P$ values provided by limma for differential expression were then corrected for multiple testing using the Benjamini–Hochberg (FDR) method.

The t-statistic generated by limma was used as input for GSEA for MSigDB hallmark gene sets[37] using the R package fgsea (v.1.16.0[96]) with default parameters. This analysis was run in R v.4.0.0.

## Statistical testing

Statistical tests were performed in R (versions 4.2.2) or Prism 9.2.0. No statistical methods were used to predetermine the sample size. Details of all statistical analyses are provided within figure legends. For all statistical tests, the number of data points included are plotted or annotated in the corresponding figure, and all statistical tests were two-sided unless otherwise specified.

## Reporting summary

Further information on research design is available in the Nature Portfolio Reporting Summary linked to this article.

## Data availability

The whole-exome sequencing data (primary tumor data from the TRACERx study and PDX models data) used during this study have been deposited with the European Genome–phenome Archive (EGA), which is hosted by The European Bioinformatics Institute (EBI) and the Centre for Genomic Regulation (CRG) under study accession code EGAS00001007364 and dataset accession code EGAD00001012228. Access is controlled by the TRACERx data access committee and details regarding applications for access are available on the relevant EGA page. NSG mouse whole-genome sequencing data have been deposited with The European Nucleotide Archive (ENA) and are publicly available under the accession code PRJEB65917. The processed data, including single nucleotide polymorphisms of the NSG mouse and the NSG-adapted mouse reference genome, as well as code to reproduce these are available via Zenodo (https://doi.org/10.5281/zenodo.10304174)[69]. The GRCm38/mm10 genome assembly can be downloaded from UCSC.

Biological materials, including PDX models generated within this study, are available to the community for academic non-commercial research purposes via standard MTA agreements.

## Code availability

All code to reproduce the figures in this manuscript is available via Zenodo (https://doi.org/10.5281/zenodo.7434887)[97]. Scripts to reproduce the NSG-adapted reference genome are available via Zenodo (https://doi.org/10.5281/zenodo.10304174)[69]. Histology overview figures were generated from digital pathology images using PATHOverview (code available at https://github.com/EpiCENTR-Lab/PATHOverview)[58]. PDX growth curves and PDX lineage relationships were monitored using PDX-Tracker (code available at https://github.com/EpiCENTR-Lab/PDX-Tracker)[98].

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

## Acknowledgements
The authors thank the staff of the Advanced Sequencing Facility at The Francis Crick Institute, as well as the members of the TRACERx consortium for their contributions to this study. TRACERx (Clinicaltrials.gov no: NCT01888601) is sponsored by University College London (UCL/12/0279) and was approved by an independent research ethics committee (REC 13/LO/1546). TRACERx is funded by Cancer Research UK (CRUK; C11496/A17786) and is coordinated by CRUK and the UCL Cancer Trials Centre. The authors thank Sharon Vanloo for administrative support and members of the EpiCENTR, Swanton and McGranahan labs for their comments on draft manuscripts. Schematic diagrams were created with BioRender.com. R.E.H. was supported by a Sir Henry Wellcome Postdoctoral Fellowship (Wellcome Trust; WT209199/Z/17) and received additional funding for this project from the CRUK Lung Cancer Centre of Excellence, the Roy Castle Lung Cancer Foundation and the James Tudor Foundation. T.K. is supported by the Japan Society for the Promotion of Science (JSPS) overseas research fellowships program (202060447). PDX derivation and WES for the samples used to validate the NSG-adapted reference genome were funded through a grant from the Moulton Charitable Foundation (to P.A.J.C). M.J.H. is a CRUK Fellow and has received funding from CRUK, NIHR, the Rosetrees Trust, UKI NETs and the NIHR University College London Hospitals Biomedical Research Centre. N.M. is a Sir Henry Dale Fellow, jointly funded by the Wellcome Trust and the Royal Society (211179/Z/18/Z), and also receives funding from CRUK, the Rosetrees Trust, the NIHR BRC at University College London Hospitals and the CRUK University College London Experimental Cancer Medicine Centre. C.S. is a Royal Society Napier Research Professor (RSRP\R\210001). His work is supported by the Francis Crick Institute that receives its core funding from Cancer Research UK (CC2041), the UK Medical Research Council (CC2041), and the Wellcome Trust (CC2041). C.S. is funded by CRUK (TRACERx C11496/A17786), PEACE (C416/A21999) and CRUK Cancer Immunotherapy Catalyst Network); CRUK Lung Cancer Centre of Excellence (C11496/A30025); the Rosetrees Trust, Butterfield and Stoneygate Trusts; NovoNordisk Foundation (ID16584); Royal Society Professorship Enhancement Award (RP/EA/180007 & RF\ERE\231118)); National Institute for Health Research (NIHR) University College London Hospitals Biomedical Research Centre; the CRUK-University College London Centre; Experimental Cancer Medicine Centre; the Breast Cancer Research Foundation (US) (BCRF-22-157); CRUK Early Detection and Diagnosis Primer Award (Grant EDDPMA-Nov21/100034); and The Mark Foundation for Cancer Research Aspire Award (Grant 21-029-ASP). This work was supported by a Stand Up To Cancer (SU2C)-LUNGevity-American Lung Association Lung Cancer Interception Dream Team Translational Research Grant (Grant Number: SU2C-AACR-DT23-17 to S.M. Dubinett and A.E. Spira). Stand Up To Cancer is a division of the Entertainment Industry Foundation. Research grants are administered by the American Association for Cancer Research, the Scientific Partner of SU2C. C.S. is in receipt of an ERC Advanced Grant (PROTEUS) from the European Research Council under the European Union's Horizon 2020 research and innovation programme (grant agreement no. 835297). For the purpose of open access, the authors have applied a CC BY public copyright license to any Author Accepted Manuscript version arising from this submission.

## Author contributions
Conceptualization: R.E.H.; Project license for animal studies: S.M.J.; Experimentation: R.E.H., D.R.P., K.H.C.G., A.B.A., D.C., K.K.K., R.T., and A.J.R.; Sample management: L.B.-C., C.N.-L., A.J.R., and S.V.; Sequencing: S.W.; Bioinformatics pipeline: A.H., M.S.H., G.A.W., K.W., and D.E.C.; Bioinformatics analysis: A.H., O.P., and C.M.-R.; Validation: A.S.M.M.H. and S.P.P.; Data visualization: R.E.H., A.H., and D.R.P.; Clinical data curation: T.K. and M.A.B.; Pathology: A.U.A., D.A.M., M.S., and T.M.; Resources: P.A.J.C. and C.D.; Project administration: R.E.H.; Supervision: R.E.H., K.L., S.A.Q., T.M., N.M., and C.S.; Funding acquisition: R.E.H., M.J.-H., and C.S.; Writing— original draft: R.E.H., A.H. and D.R.P.; Writing—review and editing: R.E.H., A.H., D.R.P., K.H.C.G., E.G., N.M., and C.S.

## Competing interests
D.A.M. reports speaker fees from AstraZeneca, Eli Lilly, BMS and Takeda, consultancy fees from AstraZeneca, Thermo Fisher, Takeda, Amgen, Janssen, MIM Software, Bristol-Myers Squibb and Eli Lilly and has received educational support from Takeda and Amgen. C.S. acknowledges grants from AstraZeneca, Boehringer-Ingelheim, Bristol Myers Squibb, Pfizer, Roche-Ventana, Invitae (previously Archer Dx Inc. - collaboration in minimal residual disease sequencing technologies), Ono Pharmaceutical, and Personalis. He is chief investigator for the AZ MeRmaiD 1 and 2 clinical trials and is the steering committee chair. He is also co-chief investigator of the NHS Galleri trial funded by GRAIL and a paid member of GRAIL's Scientific Advisory Board. He receives consultant fees from Achilles Therapeutics (also a scientific advisory board (SAB) member), Bicycle Therapeutics (also a SAB member), Genentech, Medicxi, China Innovation Centre of Roche (CICoR) formerly Roche Innovation Centre – Shanghai, Saga Diagnostics SAB member, Metabomed (until July 2022), Relay Therapeutics SAB member, and the Sarah Cannon Research Institute. C.S has received honoraria from Amgen, AstraZeneca, Bristol Myers Squibb, GlaxoSmithKline, Illumina, MSD, Novartis, Pfizer, and Roche-Ventana. C.S. has previously held stock options in Apogen Biotechnologies and GRAIL, and currently has stock options in Epic Bioscience, Bicycle Therapeutics, and has stock options and is co-founder of Achilles Therapeutics. S.V. is a co-inventor to a patent of methods for detecting molecules in a sample (U.S. patent no. 10578620; Methods for detecting molecules in a sample). C.S declares a patent application (PCT/US2017/028013; Methods for lung cancer detection) for methods to lung cancer; targeting neoantigens (PCT/EP2016/059401; Method for treating cancer); identifying patent response to immune checkpoint blockade (PCT/EP2016/071471; "Immune checkpoint intervention" in cancer), determining HLA LOH (PCT/GB2018/052004; Analysis of HLA alleles in tumors and the uses thereof); predicting survival rates of patients with cancer (PCT/GB2020/050221; Method of predicting survival rates for cancer patients), identifying patients who respond to cancer treatment (PCT/GB2018/051912; Method for identifying responders to cancer treatment); methods for lung cancer detection (US20190106751A1; Methods for lung cancer detection); methods for systems and tumor monitoring (PCT/EP2022/077987; Methods and systems for tumor monitoring). C.S. is an inventor on a European patent application (PCT/GB2017/053289; Method of detecting tumor recurrence) relating to assay technology to detect tumor recurrence. This patent has been licensed to a commercial entity and under their terms of employment C.S is due a revenue share of any revenue generated from such license(s). The remaining authors declare no competing interests.

## Additional information

[1]Cancer Research UK Lung Cancer Centre of Excellence, University College London Cancer Institute, London, UK. [2]Cancer Evolution and Genome Instability Laboratory, The Francis Crick Institute, London, UK. [3]Epithelial Cell Biology in ENT Research Group (EpiCENTR), Developmental Biology and Cancer, Great Ormond Street University College London Institute of Child Health, London, UK. [4]Cancer Genome Evolution Research Group, Cancer Research UK Lung Cancer Centre of Excellence, University College London Cancer Institute, London, UK. [5]Department of Cellular Pathology, University College London Hospitals, London, UK. [6]Advanced Sequencing Facility, The Francis Crick Institute, London, UK. [7]Lungs for Living Research Centre, UCL Respiratory, University College London, London, UK. [8]Cancer Metastasis Laboratory, University College London Cancer Institute, London, UK. [9]Cancer Research UK National Biomarker Centre, University of Manchester, Manchester, UK. [10]Cancer Research UK Lung Cancer Centre of Excellence, University of Manchester, Manchester, UK. [11]Cancer Immunology Unit, Research Department of Haematology, University College London Cancer Institute, London, UK. [12]Biological Services Unit, University College London, London, UK. [13]School of Medicine, University of Leeds, Leeds, UK. [14]Tumour Immunogenomics and Immuno-surveillance Laboratory, University College London Cancer Institute, London, UK. [15]Division of Infection, Immunity and Respiratory Medicine, University of Manchester, Manchester, UK. [16]Department of Oncology, University College London Hospitals, London, UK. [108]These authors contributed equally: Robert E. Hynds, Ariana Huebner, David R. Pearce. ✉e-mail: rob.hynds@ucl.ac.uk; nicholas.mcgranahan.10@ucl.ac.uk; charles.swanton@crick.ac.uk

## TRACERx consortium

Charles Swanton [1,2,16]✉, Nicholas McGranahan [1,4]✉, Robert E. Hynds [1,2,3,108]✉, Ariana Huebner [1,2,4,108], David R. Pearce [1,2,108], Mark S. Hill [2], David A. Moore [1,2,5], Sophia Ward [1,2,6], Takahiro Karasaki [1,2,8], Maise Al Bakir [1,2], Gareth A. Wilson [2], Oriol Pich [1,2], Carlos Martínez-Ruiz [1,4], Monica Sivakumar [1,5], Eva Grönroos [2], Cristina Naceur-Lombardelli [1], Andrew J. Rowan [2], Selvaraju Veeriah [1], Kevin Litchfield [1,14], Philip A. J. Crosbie [10,15], Caroline Dive [9,10], Sergio A. Quezada [1,11], Sam M. Janes [7], Mariam Jamal-Hanjani [1,8,16], Teresa Marafioti [5], Jason F. Lester [17], Amrita Bajaj [18], Apostolos Nakas [18], Azmina Sodha-Ramdeen [18], Mohamad Tufail [18], Molly Scotland [18], Rebecca Boyles [18], Sridhar Rathinam [18], Claire Wilson [19], Domenic Marrone [20], Sean Dulloo [18,20], Dean A. Fennell [18,20], Gurdeep Matharu [21], Jacqui A. Shaw [21], Ekaterini Boleti [22], Heather Cheyne [23], Mohammed Khalil [23], Shirley Richardson [23], Tracey Cruickshank [23], Gillian Price [24,25], Keith M. Kerr [25,26], Sarah Benafif [16,27], Jack French [27], Kayleigh Gilbert [27], Babu Naidu [28], Akshay J. Patel [29], Aya Osman [30], Carol Enstone [30], Gerald Langman [30], Helen Shackleford [30], Madava Djearaman [30], Salma Kadiri [30], Gary Middleton [30,31], Angela Leek [32], Jack Davies Hodgkinson [32], Nicola Totton [32], Angeles Montero [33], Elaine Smith [33], Eustace Fontaine [33], Felice Granato [33], Antonio Paiva-Correia [34], Juliette Novasio [33], Kendadai Rammohan [33], Leena Joseph [33], Paul Bishop [33], Rajesh Shah [33], Stuart Moss [33], Vijay Joshi [33], Katherine D. Brown [10,35], Mathew Carter [10,35], Anshuman Chaturvedi [10,35], Pedro Oliveira [10,35], Colin R. Lindsay [10,36], Fiona H. Blackhall [10,36], Matthew G. Krebs [36], Yvonne Summers [10,36], Alexandra Clipson [9,10], Jonathan Tugwood [9,10], Alastair Kerr [9,10], Dominic G. Rothwell [9,10], Hugo J. W. L. Aerts [37,38,39], Roland F. Schwarz [40,41], Tom L. Kaufmann [41,42], Rachel Rosenthal [2], Peter Van Loo [43,44,45], Nicolai J. Birkbak [1,2,46,47,48], Zoltan Szallasi [49,50,51], Judit Kisistok [46,47,48], Mateo Sokac [46,47,48], Roberto Salgado [52,53], Miklos Diossy [49,50,54], Jonas Demeulemeester [55,56,57], Abigail Bunkum [1,8,58], Angela Dwornik [59], Alastair Magness [60], Alexander M. Frankell [1,2], Angeliki Karamani [59], Antonia Toncheva [1], Benny Chain [59], Carla Castignani [45,61], Chris Bailey [2], Christopher Abbosh [1], Clare Puttick [1,2,4], Clare E. Weeden [60], Claudia Lee [2], Corentin Richard [1], Crispin T. Hiley [1,2], Despoina Karagianni [59], Dhruva Biswas [1,2,62], Dina Levi [60], Elizabeth Larose Cadieux [45,61], Emilia L. Lim [1,2], Emma Colliver [2], Emma Nye [63], Felipe Gálvez-Cancino [59], Francisco Gimeno-Valiente [1], George Kassiotis [60,64], Georgia Stavrou [59], Gerasimos-Theodoros Mastrokalos [59], Helen L. Lowe [59], Ignacio Garcia Matos [59], Imran Noorani [60], Jacki Goldman [60], James L. Reading [59], James R. M. Black [1,4], Jayant K. Rane [2,59], Jerome Nicod [6], John A. Hartley [59], Karl S. Peggs [11,65], Katey S. S. Enfield [2], Kayalvizhi Selvaraju [59], Kerstin Thol [1,4], Kevin W. Ng [66], Kezhong Chen [59], Krijn Dijkstra [60], Kristiana Grigoriadis [1,2,4], Krupa Thakkar [1], Leah Ensell [59], Mansi Shah [59], Maria Litovchenko [59],

Mariana Werner Sunderland[1], Matthew R. Huska[67], Michelle Dietzen[1,2,4], Michelle M. Leung[1,2,4], Mickael Escudero[60], Mihaela Angelova[2], Miljana Tanić[61,68], Nnennaya Kanu[1], Olga Chervova[59,69], Olivia Lucas[1,2,58,70], Othman Al-Sawaf[59,71], Paulina Prymas[1], Philip Hobson[60], Piotr Pawlik[59], Richard Kevin Stone[63], Robert Bentham[1,4], Roberto Vendramin[1,2,14], Sadegh Saghafinia[1], Samuel Gamble[59], Seng Kuong Anakin Ung[59], Sharon Vanloo[1], Simone Zaccaria[1,58], Sonya Hessey[1,8,58], Sian Harries[1,2,6], Stefan Boeing[60], Stephan Beck[61], Supreet Kaur Bola[59], Tamara Denner[60], Thomas B. K. Watkins[2,59], Thomas Patrick Jones[4], Victoria Spanswick[59], Vittorio Barbè[60], Wei-Ting Lu[60], William Hill[60], Wing Kin Liu[1,8], Yin Wu[59], Yutaka Naito[60], Zoe Ramsden[60], Catarina Veiga[72], Gary Royle[73], Charles-Antoine Collins-Fekete[74], Francesco Fraioli[75], Paul Ashford[76], Martin D. Forster[1,16], Siow Ming Lee[1,16], Elaine Borg[5], Mary Falzon[5], Dionysis Papadatos-Pastos[16], James Wilson[16], Tanya Ahmad[16], Alexander James Procter[77], Asia Ahmed[77], Magali N. Taylor[77], Arjun Nair[77,78], David Lawrence[79], Davide Patrini[79], Neal Navani[7,80], Ricky M. Thakrar[7,80], Emilie Martinoni Hoogenboom[70], Fleur Monk[70], James W. Holding[70], Junaid Choudhary[70], Kunal Bhakhri[70], Marco Scarci[70], Pat Gorman[70], Reena Khiroya[5], Robert C. M. Stephens[70], Yien Ning Sophia Wong[70], Zoltan Kaplar[81,82], Steve Bandula[70], Allan Hackshaw[83], Anne-Marie Hacker[83], Abigail Sharp[83], Sean Smith[83], Harjot Kaur Dhanda[83], Camilla Pilotti[83], Rachel Leslie[83], Anca Grapa[84], Hanyun Zhang[84], Khalid AbdulJabbar[85], Xiaoxi Pan[86], Yinyin Yuan[86], David Chuter[87], Mairead MacKenzie[87], Serena Chee[88], Aiman Alzetani[88], Judith Cave[89], Jennifer Richards[88], Eric Lim[90,91], Paulo De Sousa[91], Simon Jordan[91], Alexandra Rice[91], Hilgardt Raubenheimer[91], Harshil Bhayani[91], Lyn Ambrose[91], Anand Devaraj[91], Hema Chavan[91], Sofina Begum[91], Silviu I. Buderi[91], Daniel Kaniu[91], Mpho Malima[91], Sarah Booth[91], Andrew G. Nicholson[91,92], Nadia Fernandes[91], Pratibha Shah[91], Chiara Proli[91], Madeleine Hewish[93,94], Sarah Danson[95,96], Michael J. Shackcloth[97], Lily Robinson[98], Peter Russell[98], Kevin G. Blyth[99,100,101], Andrew Kidd[102], Craig Dick[103], John Le Quesne[104,105,106], Alan Kirk[107], Mo Asif[107], Rocco Bilancia[107], Nikos Kostoulas[107] & Mathew Thomas[107]

[17]Singleton Hospital, Swansea Bay University Health Board, Swansea, UK. [18]University Hospitals of Leicester NHS Trust, Leicester, UK. [19]Leicester Medical School, University of Leicester, Leicester, UK. [20]University of Leicester, Leicester, UK. [21]Cancer Research Centre, University of Leicester, Leicester, UK. [22]Royal Free London NHS Foundation Trust, London, UK. [23]Aberdeen Royal Infirmary NHS Grampian, Aberdeen, UK. [24]Department of Medical Oncology, Aberdeen Royal Infirmary NHS Grampian, Aberdeen, UK. [25]University of Aberdeen, Aberdeen, UK. [26]Department of Pathology, Aberdeen Royal Infirmary NHS Grampian, Aberdeen, UK. [27]The Whittington Hospital NHS Trust, London, UK. [28]Birmingham Acute Care Research Group, Institute of Inflammation and Ageing, University of Birmingham, Birmingham, UK. [29]Guy's and St Thomas' NHS Foundation Trust, London, UK. [30]University Hospital Birmingham NHS Foundation Trust, Birmingham, UK. [31]Institute of Immunology and Immunotherapy, University of Birmingham, Birmingham, UK. [32]Manchester Cancer Research Centre Biobank, Manchester, UK. [33]Wythenshawe Hospital, Manchester University NHS Foundation Trust, Wythenshawe, UK. [34]Manchester University NHS Foundation Trust, Manchester, UK. [35]The Christie NHS Foundation Trust, Manchester, UK. [36]Division of Cancer Sciences, The University of Manchester and The Christie NHS Foundation Trust, Manchester, UK. [37]Artificial Intelligence in Medicine (AIM) Program, Mass General Brigham, Harvard Medical School, Boston, MA, USA. [38]Department of Radiation Oncology, Brigham and Women's Hospital, Dana-Farber Cancer Institute, Harvard Medical School, Boston, MA, USA. [39]Radiology and Nuclear Medicine, CARIM & GROW, Maastricht University, Maastricht, The Netherlands. [40]Institute for Computational Cancer Biology, Center for Integrated Oncology (CIO), Cancer Research Center Cologne Essen (CCCE), Faculty of Medicine and University Hospital Cologne, University of Cologne, Köln, Germany. [41]Berlin Institute for the Foundations of Learning and Data (BIFOLD), Berlin, Germany. [42]Berlin Institute for Medical Systems Biology, Max Delbrück Center for Molecular Medicine in the Helmholtz Association (MDC), Berlin, Germany. [43]Department of Genetics, The University of Texas MD Anderson Cancer Center, Houston, Texas, USA. [44]Department of Genomic Medicine, The University of Texas MD Anderson Cancer Center, Houston, Texas, USA. [45]Cancer Genomics Laboratory, The Francis Crick Institute, London, UK. [46]Department of Molecular Medicine, Aarhus University Hospital, Aarhus, Denmark. [47]Department of Clinical Medicine, Aarhus University, Aarhus, Denmark. [48]Bioinformatics Research Centre, Aarhus University, Aarhus, Denmark. [49]Danish Cancer Society Research Center, Copenhagen, Denmark. [50]Computational Health Informatics Program, Boston Children's Hospital, Boston, MA, USA. [51]Department of Bioinformatics, Semmelweis University, Budapest, Hungary. [52]Department of Pathology, ZAS Hospitals, Antwerp, Belgium. [53]Division of Research, Peter MacCallum Cancer Centre, Melbourne, Australia. [54]Department of Physics of Complex Systems, ELTE Eötvös Loránd University, Budapest, Hungary. [55]Integrative Cancer Genomics Laboratory, VIB Center for Cancer Biology, Leuven, Belgium. [56]VIB Center for AI & Computational Biology, Leuven, Belgium. [57]Department of Oncology, KU Leuven, Leuven, Belgium. [58]Computational Cancer Genomics Research Group, University College London Cancer Institute, London, UK. [59]University College London Cancer Institute, London, UK. [60]The Francis Crick Institute, London, UK. [61]Medical Genomics, University College London Cancer Institute, London, UK. [62]Bill Lyons Informatics Centre, University College London Cancer Institute, London, UK. [63]Experimental Histopathology, The Francis Crick Institute, London, UK. [64]Department of Infectious Disease, Faculty of Medicine, Imperial College London, London, UK. [65]Department of Haematology, University College London Hospitals, London, UK. [66]Retroviral Immunology Group, The Francis Crick Institute, London, UK. [67]Bioinformatics and Systems Biology, Method Development and Research Infrastructure, Robert Koch Institute, Nordufer 20, 13353 Berlin, Germany. [68]Experimental Oncology, Institute for Oncology and Radiology of Serbia, Belgrade, Serbia. [69]University College London Department of Epidemiology and Health Care, London, UK. [70]University College London Hospitals, London, UK. [71]Department I of Internal Medicine, University Hospital of Cologne, Cologne, Germany. [72]Centre for Medical Image Computing, Department of Medical Physics and Biomedical Engineering, University College London, London, UK. [73]Department of Medical Physics and Bioengineering, University College London Cancer Institute, London, UK. [74]Department of Medical Physics and Biomedical Engineering, University College London, London, UK. [75]Institute of Nuclear Medicine, Division of Medicine, University College London, London, UK. [76]Institute of Structural and Molecular Biology, University College London, London, UK. [77]Department of Radiology, University College London Hospitals, London, UK. [78]UCL Respiratory, Department of Medicine, University College London, London, UK. [79]Department of Thoracic Surgery, University College London Hospital NHS Trust, London, UK. [80]Department of Thoracic Medicine, University College London Hospitals, London, UK. [81]Integrated Radiology Department, North-buda St. John's Central Hospital, Budapest, Hungary. [82]Institute of Nuclear Medicine, University College London Hospitals, London, UK. [83]Cancer Research UK & UCL Cancer Trials Centre, London, UK. [84]The Institute of Cancer Research, London, UK. [85]Case45, London, UK. [86]The University of Texas MD Anderson Cancer Center, Houston, USA. [87]Independent Cancer Patients' Voice, London, UK. [88]University Hospital Southampton NHS Foundation Trust, Southampton, UK. [89]Department of Oncology, University Hospital Southampton NHS Foundation Trust, Southampton, UK. [90]Academic

Division of Thoracic Surgery, Imperial College London, London, UK. [91]Royal Brompton and Harefield Hospitals, part of Guy's and St Thomas' NHS Foundation Trust, London, UK. [92]National Heart and Lung Institute, Imperial College, London, UK. [93]Royal Surrey Hospital, Royal Surrey Hospitals NHS Foundation Trust, Guildford, UK. [94]University of Surrey, Guildford, UK. [95]University of Sheffield, Sheffield, UK. [96]Sheffield Teaching Hospitals NHS Foundation Trust, Sheffield, UK. [97]Liverpool Heart and Chest Hospital, Liverpool, UK. [98]Princess Alexandra Hospital, The Princess Alexandra Hospital NHS Trust, Harlow, UK. [99]School of Cancer Sciences, University of Glasgow, Glasgow, UK. [100]Beatson Institute for Cancer Research, University of Glasgow, Glasgow, UK. [101]Queen Elizabeth University Hospital, Glasgow, UK. [102]Institute of Infection, Immunity & Inflammation, University of Glasgow, Glasgow, UK. [103]NHS Greater Glasgow and Clyde, Glasgow, UK. [104]Cancer Research UK Scotland Institute, Glasgow, UK. [105]Institute of Cancer Sciences, University of Glasgow, Glasgow, UK. [106]NHS Greater Glasgow and Clyde Pathology Department, Queen Elizabeth University Hospital, Glasgow, UK. [107]Golden Jubilee National Hospital, Clydebank, UK.

