## [Peer Review File · Nature Communications]

Representation of genomic intratumor heterogeneity in multi-region non-small cell lung cancer patient-derived xenograft modelsREVIEWER COMMENTS

Reviewer #1 (Remarks to the Author): expertise in non-small cell lung cancer genomics

In this study, Hynds and colleagues establish certain limitations of single PDX models generated from primary patient non-small cell lung tumors. Through whole exome sequencing of PDXs and their matched patient tumors, the authors demonstrate that PDX model establishment was a genomic bottleneck event, with ~76% of PDX models derived from a single primary tumor subclone. They show that there is heterogeneity in the ability of patient tumors to form PDXs which is dependent on several factors such as T-cell infiltration and tumor purity. Moreover, the results from their studies indicate that most PDXs are monoclonal with respect to the patient tumor, and the few that are polyclonal become restricted to single clonality during the passaging of tumors in NSG mice. Their data indicate that PDX libraries derived from at least 2 tumor regions can capture intratumor heterogeneity. Through these studies, the authors establish the importance of generating PDXs from multiple regions of primary lung tumors, which may capture the heterogeneities present to a greater extent, and which will be critical for the success of therapy resistance studies and personalized medicine approaches. Overall, this study is clearly described, and the manuscript is well-written. However, there are several concerns that need to be addressed to merit publication in Nature Communications. Major and minor comments are included below:

Major Comments:

1. The conclusions from Figure 1E are not clear. Does PDX engraftment efficiency correlate somehow to disease-free survival? The authors should clarify that this trend was not significant.
2. Sup Fig 6G shows increased T cell infiltration for primary tumors that had no engraftment (no PDX) compared to PDXs. It has been suggested that the resultant decrease in viability of the tumor cells is partly responsible for the diminished engraftment of these tumors. Are there viability differences between the PDX vs no PDX tumor samples, has this been tested in vivo or in vitro?
3. In Fig 2, Sup Fig 6E & 6H, it has been suggested that the engraftment issues of some primary tumors could be associated with decreased mutational status for TP53 and loss of heterozygosity. How do you think these factors play a role in engraftment? Is there any evidence to prove this hypothesis?
4. In Fig 4B, for the establishment of monoclonal or polyclonal PDX engraftment, what were the observations for the PDXs formed from multiple regions of the primary patient tumor?
5. In Fig 4E, it is not clear how the maximum cancer cell fraction (CCF) was calculated. The authors should more clearly describe this methodology. Can other validation studies be performed to confirm CCF other than WES?
6. In Fig 5B, some form of graphical representation in addition to the current table would help better understand the differences in clonality of PDX and primary tumor.
7. Can the authors provide any additional evidence to validate genome evolution during PDX expansion?
8. A less provocative title is recommended for this manuscript. 'Fail' is perhaps too strong a word for the evidence that is provided. It is also suggested that the tumor type 'non-small cell lung cancer' be included in the title.

Minor Comments:

1. In Figure 1D, the 'n' of PDXs used for this analysis is not mentioned in the figure legend.
2. Sample purity is not explained in Sup Fig 6F.
3. In Fig 4, the 'n' of PDXs for each analysis is missing.
4. In Sup Fig 9E, the allele frequency maps are not explained.
5. Line 315, consistent "with"

Reviewer #2 (Remarks to the Author): expertise in lung cancer PDX model development

The authors described the establishment of PDX from 145 multi-regions collections from 44 NSCLC patient (22 adeno, 16 squamous, 6 others, counted from Figure 1B) tumors, generating 63 xenografts. The final success rate, excluding lymphocytic tumors was 47 PDX derived from 22 patient tumors (7 adeno, 10 squamous, 5 others), with a final 44 PDX WES to compare against 22 paired patient tumors (Fig.1). Nine patients had PDX models derived from multiple regions (region range 2-6, median 4) of the individual primary tumors. 22 Patient tumors that generated models and those that failed to generate models were summarized by their mutations as truncal and subclonal, by copy number, selected SBS signatures, and driver genes as defined by 20 most frequently mutated genes in this cohort. It is unclear how many multi-region tumors contributed to the patient level summary (Fig.2). Using their derived "mutational distance score" and "copy number distance" to represent similarity/diversity of clonal make up in the PDX vs matched primary vs patients in TRACRx421, the authors concluded that PDX establishment was a genomic bottleneck event, with 76% of PDX models being derived from a single primary tumor subclone. By analysing PDX models that have undergone multiple passages, the authors concluded that acquisition of somatic mutations continued during PDX model expansion and was associated with APOBEC- or mismatch repair deficiency-induced mutational signatures in one individual PDX models. Based on these findings, the authors concluded that overall, while PDX models retained truncal genomic alterations that are present in all cells within the primary tumor, the failure to capture subclonal heterogeneity representative of the primary tumor is a major limitation, particularly for studies of therapy resistance and for personalized medicine approaches.

General comments: This paper provides some interesting data that incrementally increase our knowledge on NSCLC PDX. However, the evidence for supporting the conclusion (the failure to capture subclonal heterogeneity representative of the primary tumor is a major limitation, particularly for studies of therapy resistance and for personalized medicine approaches) is inadequate. More importantly, the above conclusion together with the manuscript title may give an unintended and wrongful impression to most readers who only read the abstract, that PDX is not a useful preclinical model. The small sample size of the primary patient and paired PDX samples available for analysis severely limits the strength of the data.

Specific comments:

(I). In this study, all types of NSCLC were analysed together. Knowing that LUSC and LUAD demonstrate significantly different etiologies, histological features (e.g., necrosis), genomic aberrations, transcriptomic and methylomic profiles, and therapeutic strategies, this approach is no longer acceptable. These reasons should also apply to analyses of PDX models.

(II). Large number of manuscripts on NSCLC PDX model generation have consistently reported: (1) PDX establishment rate is lower for LUAD than LUSC, (2) the ability for early stage resected NSCLC to form PDX is a poor prognostic marker for the patient, (3) most PDX models carry the truncal driver mutations found in patient tumors, and (4) PDX histology mostly resembles that of the patient tumor. The data from this study are largely consistent with these findings, although the authors somewhat disagreed with two of them (1 and 4).

- With regards to the different engraftment rates for LUAD and LUSC, the authors suggested that this may not be true when regional sampling is considered, that tumor sampling and absolute viable tumor cell number injected is more likely the determinant of engraftment rate regardless of histology type. However, their engraftment rate for LUAD was consistent with and similar to those reported in other studies, and the reduced engraftment rate when calculated by region was only observed in LUSC but not in LUAD. It should be pointed out that the 50-60% engraftment rates in LUSC have been reported by many groups that have implanted large number (>100) patient tumors, while the cohort size in this study is only 44 (24 LUAD, 16 LUSC). This suggests that the inconsistency is more likely related to the small sample size in this study. Importantly, the sample purity data (Suppl. Figure 6F) and T-cell fraction data that were used to support their suggestion were assessed using all cohort, instead of separate assessment in LUAD and LUSC, contrary to the observation that only LUSC engraftment rate was different (lower) than in other studies.

- With regards to histology, variations were noted in some cases. The authors chose to highlight these variations descriptively using terms that are commonly used by pathologists (e.g., epithelioid, rhaboid, clear cell, etc), yet likely difficult to be understood by non-pathologists or are

interpreted differently among pathologists. The evidence would have been stronger if quantitative data is provided and if it is supported by transcriptomic data that show gene expression divergence.

(III). Aside from the consistent findings outlined above, many outstanding questions involving PDX and parental (patient) tumors remain. Two of these issues, clonal representation of PDX on the patient tumor at multi-omics levels and genomic and clonal evolution of PDX with increasing passages, are partly addressed by this manuscript, but only using genomics (mutation and copy number, both data types derived from whole exome data) rather than in multi-omics level. However, there are other critical issues, such as (a) omic factors that determine PDX engraftment rate, (b) molecular basis of its poor prognostic association, (c) predictiveness of PDX drug response profile for patient tumor at recurrence, (d) predictiveness of PDX tumor from advanced patients to therapy, (e) association of genomic evolution with drug response. These complex issues would greatly benefit from additional in-depth analysis from data generated in TRACERx study. In the absence of such data, clonal heterogeneity by mutations and copy number alone is insufficient as basis for making a broad statement stated in the conclusion, as previously mentioned.

- The author demonstrated that clonality of regions of the primary tumor may or may not be reflected in the PDX models, with more models being monoclonal than polyclonal. Rather than this bottleneck being an undesirable biology of the PDX models, it may suggest that the monoclonal clones exert greater ability to grow in the PDX microenvironment. Considering that PDX growth is consistently associated with poor prognosis, further analysis to identify omic features of the dominant clones could potentially give more novel insight on issues outlined above. Moreover, as some TRACERx patients might have had biopsies taken from recurrent tumors for molecular analysis and PDX engraftment, their co-analyses would have provided more novel and insightful data.

- The authors showed that clonal representation of PDX at P0 is inconsistent between regions, and this is further confounded at subsequent passages in the mice. The authors are particularly concerned that this "haphazard" representation would result in the non-representation of PDX models for the primary tumor to therapy in the context of personalized medicine approach. However, data to support the belief that subclonal mutations can be used to direct therapy at resistance is not provided and hypothetical. This again is not supportive of the conclusion statement that "the failure to capture subclonal heterogeneity representative of the primary tumor is a major limitation, particularly for studies of therapy resistance and for personalized medicine approaches."

- While clonal and subclonal analysis and tracing provide very interesting computational model for tumor cell growth and evolution during PDX establishment and passages, the results remain hypothetical without biological validations using techniques such as single cell or limiting dilution experiments with bar coding.

(IV). The abstract emphasizes that "Acquisition of somatic mutations continued during PDX model expansion and was associated with APOBEC- or mismatch repair deficiency-induced mutational signatures in individual PDX models". However, evidence for APOBEC signature was mentioned in only one model (CRUK0995 R1) and mismatched repair (MMR) signature in CRUK0935. There should be more than one case each for these signatures to be cited as common mechanisms for ongoing mutational evolution in PDX during passages.

Furthermore, based on the above finding that tumor-intrinsic mutational signatures continue in some PDX models, the authors cautioned against the use of over-passaged PDX models that might have accumulated significant numbers of these mutations. Although this is a reasonable advice, the caution should be more specific as to the purpose of the study for using the models, since it remains unclear how fast and widespread genetic drift occurs in PDX, and what its impact is on the tumor biology. Furthermore, while the concept of using PDX as avatar for individual patient treatment remains controversial, PDXs are valuable preclinical models for drug studies and tumor biology research.

(V). The authors often referred to subclonal driver mutations, yet the definition of "driver

mutations" in this context is unclear. For example, Supplementary Figure 13 shows many genes listed as "driver mutations" (e.g., BHMT2, ADCY2, MGA, RIMS2, TNFR) yet the reason they are considered "drivers" is not provided. For example, BHMT2 is betaine-homocysteine S-methyltransferase 2 that plays critical role in methylation reactions. GeneCards mention that diseases associated with BHMT2 include paraneoplastic polyneuropathy and neural tube defects. Additional annotation as Census gene or Hallmark gene of BHMT2 is lacking from COSMIC. The reason for including it as a driver mutation in cancer including lung cancer needs clearer explanation or relevant citation.

(VI). Additional issues:

- The sentence "... than to spatially distinct tumor regions within the same tumor with the distance to non-region of origin comparable to that between primary tumor regions ..." (line 230-232) is difficult to understand.

- The term "homozygous" and "heterozygous" are used in the text (line 222-223) and as annotation of various colors in supplementary Figure 9A. However, their explanations are not provided in both the text and the figure legend, thus their meanings are unclear.

- In Supplementary Figure 10, black represents shared clusters. It is unclear why black is present in the phylogenetic trees but is not represented in the clone maps. The explanation of these plots in the figure legend is minimal. The figures may be understood better by genomic experts, but for most readers, more detail explanation in non-expert language should be provided.

- Line 644: the authors mentioned Copy number distance, they gave the description, but did not provide the detail of calculation, how to get the copy number distance between two regions. Are they similar with mutation distance calculation?

- New NSG-adapted reference genome was concluded to decrease contaminating mouse reads when generating the cancer cell fraction (CCF) and phylogenetic trees and comparisons between primary region and PDX region, it is unclear how much the results would be significantly affected if only the standard murine genome (mm10) was used.

Reviewer #3 (Remarks to the Author): expert in PDX development and bioinformatics analysis

In this manuscript, Hynds et al. derived PDX models from multiple regions of primary NSCLC from patients enrolled in the TRACERx study, and explored the histological and genetic fidelity of these PDX models. They performed WES on the PDX models and compared PDXs to their primary tumors and to other PDXs derived from different regions of the same tumors. Using this approach, the authors tackled several questions: (1) The extent of genomic bottleneck upon engraftment; (2) The reproducibility of PDX derivation across spatially distinct samples from the same primary tumor; and (3) the genetic stability of PDX models over in vivo passaging. In addition, the manuscript reports an artifact in reads alignment and presents an NSG-adapted approach that improves the accuracy of sequencing analyses of PDX models.

This is an interesting study that expands the utility of the TRACERx project to address important questions related to PDX modeling. The experiments and the analyses are carried out well, and the manuscript is written clearly. The conclusions are thoughtful, and the paper will be of high value for those working with PDX models. I therefore support its publication in Nature Communications. I do have a few suggestions that would strengthen the paper in my mind, and I'd encourage the authors to consider them:

Major comments

- (1) The authors demonstrate the need for an NSG-adapted mouse reference genome. This is an important observation. I assume that the authors deposited the WGS data from the NSG mice in a publicly-available repository that can be downloadable without restrictions; please confirm. (a) Can the authors also provide a table with the ~7M SNPs that they identified to be different from the mm10 reference? Providing the code that spikes in the divergent NSG SNPs to the mm10

reference to improve filtering would also be very useful for the field. (b) Can the authors analyze previous large-scale studies (e.g., Sun et al Nat Commun 2021) to assess to what extent artifact mutations shown in Figure 3 affected previous analyses?

(2) The manuscript focuses almost entirely on point mutations. Given that (a) copy number alterations (CNAs) can be easily extracted from WES data; (b) that the Swanton lab has a long-standing interest and expertise in chromosomal instability and CNAs; and (c) that previous PDX analyses have focused on CNA evolution (refs 9-11) – this aspect of the paper should be further developed. Currently, the authors show in Supplementary Fig. 9 that there is high similarity between the mutation distance and the CNA distance of the PDXs and provide an example for CNA diversity across regions (and they also address CNAs in Supplementary Fig. 12). However, each of the Main Figures of the paper could benefit from an addition of a CNA-based analysis (alongside the mutation-based analysis). This would also allow for a direct comparison of the results to those from previous studies.

For example, previous studies reported ~10%-20% CNA dissimilarity between primary tumors and PDXs, with high variance (both highly-similar and highly-divergent models). Is this the same range of changes observed in the current PDX cohort? (In this regard, it will be helpful to use additional measures for CNA comparison in addition to MSAI, such as the fraction of the genome that is CNA-concordant.)

Also, are these differences mostly the outcome of the subclonal CNA landscapes across tumor regions, as observed for mutations? Or would genomic evolution throughout propagation affect CNAs more than point mutations (e.g., due to potential negative selection against specific chromosome-scale alterations; or due to the p53 inactivation and chromosome instability of the PDXs)? All these questions can be addressed with the available data and are within the authors' expertise.

(3) When studying the genomic evolution of the PDXs, the authors compare primary-P3 samples and P0-P3 samples, reaching the conclusion that considerable genomic evolution can sometimes be observed, but most of the genomic divergence stems from the engraftment bottleneck (consistent with previous reports). It will be interesting to add a comparison of primary-P3 samples, to assess whether the genomic evolution of PDX models would make them diverge from the primary tumor throughout passaging. This is of course expected given the results that the clonal diversity decreases with passage, but it should be formally shown. (The distances between primary-P0 and P0-P3 are not necessarily additive.)

(4) Related to the previous point, the study compares P0 and P3 PDX samples. Many PDX cohorts comprise samples from much later passages, where genomic evolution might play a larger role. In case the authors have such matched late-passage PDXs available, it would be valuable to characterize those and subject them to the same types of analyses, as I'm not sure that P3 really represents a "late" passage. If no such examples are available, it's worth mentioning this point as a caveat in the Discussion.

RESPONSE TO REVIEWERS' COMMENTS

We thank the reviewers for taking the time to provide valuable feedback on our manuscript. We address their comments point-by-point below (reviewer comments in bold font).

Reviewer #1 (Remarks to the Author): expertise in non-small cell lung cancer genomics

In this study, Hynds and colleagues establish certain limitations of single PDX models generated from primary patient non-small cell lung tumors. Through whole exome sequencing of PDXs and their matched patient tumors, the authors demonstrate that PDX model establishment was a genomic bottleneck event, with ~76% of PDX models derived from a single primary tumor subclone. They show that there is heterogeneity in the ability of patient tumors to form PDXs which is dependent on several factors such as T-cell infiltration and tumor purity. Moreover, the results from their studies indicate that most PDXs are monoclonal with respect to the patient tumor, and the few that are polyclonal become restricted to single clonality during the passaging of tumors in NSG mice. Their data indicate that PDX libraries derived from at least 2 tumor regions can capture intratumor heterogeneity. Through these studies, the authors establish the importance of generating PDXs from multiple regions of primary lung tumors, which may capture the heterogeneities present to a greater extent, and which will be critical for the success of therapy resistance studies and personalized medicine approaches. Overall, this study is clearly described, and the manuscript is well-written. However, there are several concerns that need to be addressed to merit publication in Nature Communications. Major and minor comments are included below:

We thank the reviewer for their enthusiasm for our manuscript and constructive suggestions for improvements.

Major Comments:

1.1. The conclusions from Figure 1E are not clear. Does PDX engraftment efficiency correlate somehow to disease-free survival? The authors should clarify that this trend was not significant.

To clarify this, the revised manuscript now reads: "Consistent with previous reports suggesting that PDX establishment is linked to poor prognosis in NSCLC^{18,25-27}, we observed a trend towards shorter disease-free survival in patients for whom at least one PDX model was established (Log rank test, $p = 0.098$; Figure 1E)."

1.2. Sup Fig 6G shows increased T cell infiltration for primary tumors that had no engraftment (no PDX) compared to PDXs. It has been suggested that the resultant decrease in viability of the tumor cells is partly responsible for the diminished engraftment of these tumors. Are there viability differences between the PDX vs no PDX tumor samples, has this been tested in vivo or in vitro?

We find that T cell infiltration of LUSC regions negatively correlates with PDX engraftment. On the one hand, this might be caused by lower overall number of tumor cells in regions with higher stromal content, i.e. this is an indirect association. However, it might also be the case that T cells actively reduce the viability of the tumor cells. Unfortunately, the regional tumor samples received for PDX generation within our study were small and therefore we did not systematically perform total cell counts or assess viability of the injected samples in order to maximize the tissue available for engraftment. However, during the revision we have been

able to assess this using RNA sequencing data from a subset of primary tumor regions. Supplementary Figure 12C shows that both hypoxia and apoptosis-associated pathways are enriched in regions that failed to engraft in PDX models.

The revised manuscript reads:

“T cell infiltration of the primary tumor regions was lower for engrafted regions as estimated using the T cell ExTRECT tool³⁵ (overall $p = 0.028$, Wilcoxon rank sum test), and this was driven by LUSC tumors where T cell abundance was associated with failure for PDX models to engraft ($p = 0.015$, Wilcoxon rank sum test; Supplementary Figure 12B). Increased T cell content might reflect a higher stroma:tumor ratio within a region, or may directly reduce the viability of tumor cells. Consistent with the latter, analysis of RNA sequencing data from the subset of regions where data were available³⁶ found enrichment for apoptosis-related pathways (apoptosis Hallmark pathway, FDR $q = 5 \times 10^{-4}$) in regions that fail to engraft and for proliferation-related pathways in regions that engrafted (E2F targets, MYC targets v1 and G2M checkpoint Hallmark pathways, FDR $q = 1.1 \times 10^{-7}$, 2×10^{-6} and 2.1×10^{-5} , respectively; Supplementary Figure 12C).”

1.3. In Fig 2, Sup Fig 6E & 6H, it has been suggested that the engraftment issues of some primary tumors could be associated with decreased mutational status for TP53 and loss of heterozygosity. How do you think these factors play a role in engraftment? Is there any evidence to prove this hypothesis?

In the revised manuscript we have looked at these issues split by histology in response to Reviewer 2. Interestingly, we find that the effect is driven by PDX failure in a subset of LUAD patients (Figure 2; Supplementary Figure 10). 6/13 LUAD patients with *TP53* mutations engrafted compared to 1/11 LUAD patients with no detected *TP53* mutation. Both the total proportion of the genome that is aberrant and loss of heterozygosity are significantly higher in *TP53*-mutant LUAD tumors, leading us to speculate that there might be a minimal threshold of genomic instability required for initiation of LUAD PDX models, that some tumors do not reach. As it promotes genomic instability, *TP53* mutation is likely to increase the probability of reaching such a threshold.

1.4. In Fig 4B, for the establishment of monoclonal or polyclonal PDX engraftment, what were the observations for the PDXs formed from multiple regions of the primary patient tumor?

Figure 4B shows the clonality status (monoclonal or polyclonal) for all unique spatial regions for which we obtained a P0 PDX model in our cohort. In 6/9 patients with multi-region PDX models, we observed that initial P0 PDX models derived from the same tumor could be either monoclonal or polyclonal (Figure 4B). This is also depicted graphically in the context of the primary tumor sequencing data in Figure 5.

We have revised the results text to read: “Although the majority of tumor region-specific PDX models were themselves heterogeneous (40/42; 95.2%), due to the dominant monoclonal engraftment patterns observed they often represented only a single branch of the overall tumor phylogeny. In 8 of 9 cases in which we could compare multiple PDX models from the same tumor, we observed engraftment of more than one tumor subclone in independent PDX models (Figure 5).”

1.5. In Fig 4E, it is not clear how the maximum cancer cell fraction (CCF) was

calculated. The authors should more clearly describe this methodology. Can other validation studies be performed to confirm CCF other than WES?

In the revised manuscript, we use subclonal proportions rather than maximum CCF. This is described in the “Calculating clone proportions” section of the revised methods. We are not aware of independent methods that we could use to derive CCF.

1.6. In Fig 5B, some form of graphical representation in addition to the current table would help better understand the differences in clonality of PDX and primary tumor.

With the revised definition of driver mutations used in the updated manuscript (a more conservative definition is used, please see Reviewer #2 comment 2.13), we find three patients where one model has a driver and another model from the same patient does not, so we chose to summarize these in the text. The revised, more extensive table of driver mutation inconsistencies has been moved to Supplementary Figure 16.

1.7. Can the authors provide any additional evidence to validate genome evolution during PDX expansion?

We have expanded Figure 6 and Supplementary Figures 18 and 19 to include analysis of how copy number distance, the proportion of the aberrant genome and proportion of LOH change between P0 and P3. We also include a new analysis of the proportion of the genome that is identical between P0 and P3 samples (Supplementary Figure 19F).

1.8. A less provocative title is recommended for this manuscript. ‘Fail’ is perhaps too strong a word for the evidence that is provided. It is also suggested that the tumor type ‘non-small cell lung cancer’ be included in the title.

We agree with both points and have changed the manuscript title to “Representation of genomic intratumor heterogeneity in multi-region non-small cell lung cancer patient-derived xenograft models”.

Minor Comments:

1.9. In Figure 1D, the ‘n’ of PDXs used for this analysis is not mentioned in the figure legend.

The n number (40) is included in the revised Figure legend.

1.10. Sample purity is not explained in Sup Fig 6F.

We have now defined sample purity in the revised Supplementary Figure 12A figure legend.

1.11. In Fig 4, the ‘n’ of PDXs for each analysis is missing.

Apologies, the n numbers have now been added for Figure 4.

1.12. In Sup Fig 9E, the allele frequency maps are not explained.

Allele frequency maps are a visual depiction of the composition of a tumor sample, and show the proportion of cells within a tumor region that can be attributed to specific mutation clusters as defined in the phylogenetic trees. We have now added this description to both the Methods and the relevant Figure legend (revised Figure 5).

1.13. Line 315, consistent “with”

Thanks – this has been edited in the revised manuscript.

Reviewer #2 (Remarks to the Author): expertise in lung cancer PDX model

development

2.1. The authors described the establishment of PDX from 145 multi-regions collections from 44 NSCLC patient (22 adeno, 16 squamous, 6 others, counted from Figure 1B) tumors, generating 63 xenografts. The final success rate, excluding lymphocytic tumors was 47 PDX derived from 22 patient tumors (7 adeno, 10 squamous, 5 others), with a final 44 PDX WES to compare against 22 paired patient tumors (Fig.1). Nine patients had PDX models derived from multiple regions (region range 2-6, median 4) of the individual primary tumors. 22 Patient tumors that generated models and those that failed to generate models were summarized by their mutations as truncal and subclonal, by copy number, selected SBS signatures, and driver genes as defined by 20 most frequently mutated genes in this cohort. It is unclear how many multi-region tumors contributed to the patient level summary (Fig.2). Using their derived “mutational distance score” and “copy number distance” to represent similarity/diversity of clonal make up in the PDX vs matched primary vs patients in TRACRx421, the authors concluded that PDX establishment was a genomic bottleneck event, with 76% of PDX models being derived from a single primary tumor subclone. By analysing PDX models that have undergone multiple passages, the authors concluded that acquisition of somatic mutations continued during PDX model expansion and was associated with APOBEC- or mismatch repair deficiency-induced mutational signatures in one individual PDX models. Based on these findings, the authors concluded that overall, while PDX models retained truncal genomic alterations that are present in all cells within the primary tumor, the failure to capture subclonal heterogeneity representative of the primary tumor is a major limitation, particularly for studies of therapy resistance and for personalized medicine approaches.

We thank the reviewer for their summary. In Figure 2, there are 47 multi-region tumors from 44 patients. Both regions from CRUK0739 failed quality control and so this patient was excluded from all analyses apart from the cohort overview in Figure 1. For mutational analyses, a total of 215 regions from 43 patients with a median of 5 regions per tumor (range 2-8) were analyzed. For copy number analyses, a total of 208 regions from 43 patients with a median of 4 regions per tumor (range 2-8) were analyzed. We have included n numbers of patients and tumors in the revised Figure 2 legend, as well as adding the ‘n regions’ as a rug at the bottom of Figure 2.

2.2. General comments: This paper provides some interesting data that incrementally increase our knowledge on NSCLC PDX. However, the evidence for supporting the conclusion (the failure to capture subclonal heterogeneity representative of the primary tumor is a major limitation, particularly for studies of therapy resistance and for personalized medicine approaches) is inadequate. More importantly, the above conclusion together with the manuscript title may give an unintended and wrongful impression to most readers who only read the abstract, that PDX is not a useful preclinical model. The small sample size of the primary patient and paired PDX samples available for analysis severely limits the strength of the data.

We disagree that this is an incremental advance; the manuscript develops a multi-region sampling protocol for the first time in any tumor type and reveals genomic monoclonality as a significant feature of our NSCLC PDX model cohort. These findings might have implications for PDX models more generally. As our study represents a first, we do not have multi-region PDX data from other tumor types for comparison. Sample size is of course limited by the costs of running such a programme of PDX derivation.

It is not our intention to dismiss PDX models as a model system. As we state in Discussion, “The full representation of truncal tumor alterations (those present in all cells of the tumor) in

PDX models supports their use in cohort level studies and for testing therapeutics targeting truncal events". This represents a majority of current applications. We have toned down our conclusions to suggest that our data on subclonal events might have implications for the use of PDX models and that this should be investigated experimentally. As mentioned above, we have also changed the manuscript title in response to Reviewer 1's feedback.

Specific comments:

2.3. (I). In this study, all types of NSCLC were analyzed together. Knowing that LUSC and LUAD demonstrate significantly different etiologies, histological features (e.g., necrosis), genomic aberrations, transcriptomic and methylomic profiles, and therapeutic strategies, this approach is no longer acceptable. These reasons should also apply to analyses of PDX models.

In the revised manuscript, we present data for LUAD, LUSC and 'other' NSCLC histologies separately throughout the manuscript. In the clinical overview (Figure 1B), we have clarified the 'other' histologies and now present the separate clinical characteristics (Supplementary Figure 7) in the context of the histological groups. For PDX outcome modeling (Figure 1C), we present each histological group separately in Supplementary Figure 4. We also present the genomic characteristics of the patient cohort separated by histology (Figure 2; Supplementary Figures 7-12). We are grateful to the reviewer for their suggestion as this has revealed previously unappreciated differences between the histological subtypes in multiple analyses.

2.4. (II). Large number of manuscripts on NSCLC PDX model generation have consistently reported: (1) PDX establishment rate is lower for LUAD than LUSC, (2) the ability for early stage resected NSCLC to form PDX is a poor prognostic marker for the patient, (3) most PDX models carry the truncal driver mutations found in patient tumors, and (4) PDX histology mostly resembles that of the patient tumor. The data from this study are largely consistent with these findings, although the authors somewhat disagreed with two of them (1 and 4).

- With regards to the different engraftment rates for LUAD and LUSC, the authors suggested that this may not be true when regional sampling is considered, that tumor sampling and absolute viable tumor cell number injected is more likely the determinant of engraftment rate regardless of histology type. However, their engraftment rate for LUAD was consistent with and similar to those reported in other studies, and the reduced engraftment rate when calculated by region was only observed in LUSC but not in LUAD. It should be pointed out that the 50-60% engraftment rates in LUSC have been reported by many groups that have implanted large number (>100) patient tumors, while the cohort size in this study is only 44 (24 LUAD, 16 LUSC). This suggests that the inconsistency is more likely related to the small sample size in this study. Importantly, the sample purity data (Suppl. Figure 6F) and T-cell fraction data that were used to support their suggestion were assessed using all cohort, instead of separate assessment in LUAD and LUSC, contrary to the observation that only LUSC engraftment rate was different (lower) than in other studies.

We have now included a chart to put our findings into the context of the published literature (Supplementary Figure 7K). Overall, our take rate in LUAD is highly consistent with the literature regardless of whether looking at the patient or region level. In LUSC, at the patient level (n = 16) our engraftment rate is the third highest of the studies assessed when using our patient level success rate (as might be expected, as we have the advantage of multiple attempts per patient). Thus, our findings are consistent with much larger studies that report success/failure of models at the patient level based on single PDX model attempts. However, we were surprised to find our take rate at the region level was inconsistent with

this picture, with our LUSC success rate by region (n = 60) lower than other published studies (assuming sampling of a single “region” in such studies). We agree that our n number is lower than some of these studies, and that there are many potential confounding issues here (differences in biopsy type and tissue injection methods, lack of standardization of tumor cell number injected per mouse, different mouse strains etc). However, since we do see the previously reported difference at the patient-level, we believe this is an aspect of our work that should be included in the manuscript, particularly as this is the first report of multi-region PDX models in NSCLC. In the revised manuscript, we discuss this issue as follows:

“For example, prior studies suggest that lung squamous carcinomas more readily give rise to PDX than lung adenocarcinomas^{16–18,21,23,26–32}. Our patient-level engraftment rates were consistent with this, but the LUSC engraftment rate was substantially lower at the region level than the patient level (Supplementary Figure 5K). This suggests that PDX engraftment potential might be more spatially variable in LUSC tumors, although other sampling biases (e.g. higher tissue availability from larger tumors) might play a role in the apparent histology-dependent changes in engraftment rates seen in other studies and a relatively small number of LUSC tumors (n = 16 patients, 60 regions) were analyzed here.”

Consistent with the wider changes we have made to the manuscript, the purity (Supplementary Figure 12A) and T cell (Supplementary Figure 12B) data are now provided broken down by histological subtypes. Purity was significantly associated with engraftment in ‘other’ NSCLC histologies and there was a trend in LUAD and LUSC. The T cell fraction in the tumor was significantly higher in LUSC tumor regions that did not engraft than those that did with no difference observed in LUAD.

2.5. - With regards to histology, variations were noted in some cases. The authors chose to highlight these variations descriptively using terms that are commonly used by pathologists (e.g., epithelioid, rhaboid, clear cell, etc), yet likely difficult to be understood by non-pathologists or are interpreted differently among pathologists. The evidence would have been stronger if quantitative data is provided and if it is supported by transcriptomic data that show gene expression divergence.

A consultant pathologist (D.A.M.) performed a comprehensive pathology review of PDX models and their associated primary tumor regions. The results of this review are now summarized in Supplementary Figure 5A and provided as Supplementary Table 1, which we trust addresses the concern that we were overemphasizing differences by only presenting example cases.

The results are described in the revised manuscript as follows: “In a review of PDX model histopathology, we observed high consistency between initial P0 and established P3 PDX models (Supplementary Figure 5). When comparing PDX models to region-specific hematoxylin and eosin (H&E) stained sections available from patient tumors, we observed concordance for the majority of models, consistent with prior PDX models that have been shown to broadly resemble the histologies of the tumors from which they were derived^{16,16–24}. However, in a minority of cases, we noted histological variation. Some models showed evidence of divergence at P0; for example, CRUK0949 R1 and R3 showed more widespread clear cell differentiation than was present in the corresponding patient samples for those regions, and CRUK0816 R2 and R5 PDX models presented more epithelioid differentiation than the parent tumor (Supplementary Figure 6A). Other models varied between the initial P0 and established P3 samples, with the initial P0 PDX model more closely resembling the patient region than the P3 PDX model; for example, CRUK0941 R2 PDX model showed prominent rhabdoid differentiation at P3 that had not been present in either the patient or P0 samples (Supplementary Figure 6B), though this was consistent with the cytological pleomorphism seen in this poorly differentiated pleomorphic carcinoma. In multiple CRUK0606 regional PDX models, substantial variation between either tumor and P0

PDX models, or P0 and P3 PDX models was observed. Glandular features were a minor component of the patient's regional tissue but became more prominent in PDX models, either in both initial P0 and established P3 models (R5, R8) or in the established P3 model only (R1, R6; Supplementary Figure 6B)."

Our approach is described in the revised Methods section as follows: "Slides from P0 and P3 PDX models, along with region-specific H&E images from the patient tumor, were subjected to a comprehensive pathology review by a consultant pathologist. Instances in which samples were consistent with one another were scored 2 (consistent), instances in which broad similarity of histopathological subtype were observed but minor differences, for example in the prevalence of a particular growth pattern, were scored 1 (divergent), while samples that were dissimilar from one another were scored 0 (inconsistent). Differences in the extent of necrosis between samples were common but were ignored for scoring purposes as it is likely to be affected by experimental factors in addition to being a characteristic of specific tumor regions/PDX models."

Although we have bulk RNA sequencing for a subset of primary tumor regions, we do not have RNA sequencing data for our PDX models. In any case, a comparison of primary tumor and PDX bulk RNA sequencing would be difficult to interpret given the substantial differences in tumor purity between primary tumor and PDX samples.

2.6. (III). Aside from the consistent findings outlined above, many outstanding questions involving PDX and parental (patient) tumors remain. Two of these issues, clonal representation of PDX on the patient tumor at multi-omics levels and genomic and clonal evolution of PDX with increasing passages, are partly addressed by this manuscript, but only using genomics (mutation and copy number, both data types derived from whole exome data) rather than in multi-omics level. However, there are other critical issues, such as (a) omic factors that determine PDX engraftment rate, (b) molecular basis of its poor prognostic association, (c) predictiveness of PDX drug response profile for patient tumor at recurrence, (d) predictiveness of PDX tumor from advanced patients to therapy, (e) association of genomic evolution with drug response. These complex issues would greatly benefit from additional in-depth analysis from data generated in TRACERx study. In the absence of such data, clonal heterogeneity by mutations and copy number alone is insufficient as basis for making a broad statement stated in the conclusion, as previously mentioned.

We agree that many questions about NSCLC PDX models remain to be answered and these are nicely summarized by the reviewer. We will make our multi-region models available to the community to enable such studies. Multi-omic analysis of patients and PDX models is beyond the scope of the current manuscript and our three-month timeline for revision, but as noted by Reviewer 3, we do have transcriptomic data from a subset of patients within TRACERx. We have analyzed this as suggested to look for associations with PDX engraftment and find that pathways including EMT, hypoxia and apoptosis are enriched in non-engrafting tumors (Supplementary Figure 12C).

We have modified the conclusion in order to reflect the limitations of our study.

2.7. - The author demonstrated that clonality of regions of the primary tumor may or may not be reflected in the PDX models, with more models being monoclonal than polyclonal. Rather than this bottleneck being an undesirable biology of the PDX models, it may suggest that the monoclonal clones exert greater ability to grow in the PDX microenvironment. Considering that PDX growth is consistently associated with poor prognosis, further analysis to identify omic features of the dominant clones could potentially give more novel insight on issues outlined above.

Our study shows that when primary tumors contain multiple clones, only one of these is represented in most P0 PDX models (and almost all PDX models are derived from the descendants of one primary tumor clone by P3). In many multi-region PDX libraries, PDX models arise from different tumor clones in different engraftments (Figure 5), i.e. there is not reproducible selection for the same primary tumor clone in PDX libraries derived from different spatial regions of the same primary tumor. As such, our data do not strongly support the concept of the “fittest clone” (fittest in general, or fittest for mouse engraftment) of a primary tumor seeding NSCLC PDX models.

2.8. Moreover, as some TRACERx patients might have had biopsies taken from recurrent tumors for molecular analysis and PDX engraftment, their co-analyses would have provided more novel and insightful data.

We agree that TRACERx and PEACE metastasis data represents a useful resource for investigating similarities and differences between PDX models and metastases. In the revised manuscript, we have performed a combined analysis of the primary tumor regions, PDX models and metastases samples (Supplementary Figure 17). Data were available for eight such cases and while we find examples of PDX models that were seeded by the same clone as the patient metastasis, there is no clear pattern to suggest that this is frequent (with our sample size being an obvious caveat). We describe this new analysis as follows:

“For eight patients we had either matched primary lymph node or recurrence/progression WES data to investigate the similarity of PDX-engrafting clones and metastasis-seeding clones. Where the metastatic seeding clone(s) was present in the primary tumor at sampling (7/8 patients), metastatic seeding clones were found in at least one PDX model in 3/7 patients. In CRUK0640, the metastatic clone was found to engraft a monoclonal R7 P0 PDX model, as well as contributing to the engraftment of polyclonal P0 PDX models derived from R1, R5 and R8 but this clone did not seed the R3 P0 PDX model. CRUK0718 also had a single metastatic clone which was found to engraft the monoclonal P0 PDX models derived from R4 and R8, and was also found to contribute to engraftment of the polyclonal R7 P0 PDX model. CRUK0748 had multiple metastatic clones of which one was found to contribute to the engraftment of the polyclonal R6 P0 PDX model but did not engraft in the R1, R2, R3 or R8 P0 PDX models. In two further cases, the CRUK0885 R3 and CRUK0816 R2 and R5 PDX models engraftment was by the direct descendent clone of the metastasis seeding clone, although a further PDX model from CRUK0816 (R3) was engrafted by a clone on a separate branch. CRUK0941 and CRUK1061 PDX models were engrafted by clones on a distinct branch from the metastasis. Consideration of the mutational and copy number distance scores between PDX models, primary tumor regions and metastases did not strongly support the hypothesis that PDX models from the primary tumor resemble patient metastases; while in some cases, the distances between PDX models and metastases was within the range of the distances between the PDX model and the primary tumor regions, in other cases PDX models were more dissimilar to the metastases than to any primary tumor region (Supplementary Figure 17).”

2.9. - The authors showed that clonal representation of PDX at P0 is inconsistent between regions, and this is further confounded at subsequent passages in the mice. The authors are particularly concerned that this “haphazard” representation would result in the non-representation of PDX models for the primary tumor to therapy in the context of personalized medicine approach. However, data to support the belief that subclonal mutations can be used to direct therapy at resistance is not provided and hypothetical. This again is not supportive of the conclusion statement that “the failure

to capture subclonal heterogeneity representative of the primary tumor is a major limitation, particularly for studies of therapy resistance and for personalized medicine approaches."

We did not use the word haphazard in our manuscript and would not use it to describe our findings.

PDX models lacking the full complement of subclones that were present in the tumor region of origin is a cause for concern in personalized approaches where clonal selection in model systems is hoped to represent clonal selection in patients (e.g. under therapy pressure). Similarly, the ability to derive PDX models with different driver mutation profiles from the same tumor is concerning for some PDX model applications. As noted in response to reviewer 1, we have toned down our conclusions to suggest that our data on subclonal events might have implications for the use of PDX models, and that this should be investigated experimentally in future studies.

2.10. - While clonal and subclonal analysis and tracing provide very interesting computational model for tumor cell growth and evolution during PDX establishment and passages, the results remain hypothetical without biological validations using techniques such as single cell or limiting dilution experiments with bar coding.

As above, we agree that these issues should be investigated experimentally and hope that we and others will address this in the future. However, such experiments are a significant undertaking and are not achievable within the timeframe of a three-month manuscript revision.

2.11. (IV). The abstract emphasizes that "Acquisition of somatic mutations continued during PDX model expansion and was associated with APOBEC- or mismatch repair deficiency-induced mutational signatures in individual PDX models". However, evidence for APOBEC signature was mentioned in only one model (CRUK0995 R1) and mismatched repair (MMR) signature in CRUK0935. There should be more than one case each for these signatures to be cited as common mechanisms for ongoing mutational evolution in PDX during passages.

We were able to confidently observe mutational signatures in PDX models from three patients, the two mentioned above plus CRUK0748, where models had a clock-like signature. In each case, multiple independent PDX models showed evidence of the observed signatures. In the revised manuscript, we have removed these example cases from the abstract. In reality, it is likely that more models acquire mutations with characteristic mutational signatures but, as the overall number of mutations is low, these remain below the detection threshold of signature calling tools. These data are now presented in Supplementary Figure 18.

2.12. Furthermore, based on the above finding that tumor-intrinsic mutational signatures continue in some PDX models, the authors cautioned against the use of over-passaged PDX models that might have accumulated significant numbers of these mutations. Although this is a reasonable advice, the caution should be more specific as to the purpose of the study for using the models, since it remains unclear how fast and widespread genetic drift occurs in PDX, and what its impact is on the tumor biology. Furthermore, while the concept of using PDX as avatar for individual patient treatment remains controversial, PDXs are valuable preclinical models for drug studies and tumor biology research.

We agree, and have rephrased the discussion to state that: "A caveat of our study is that we did not analyze PDX tumors after P3, meaning that we cannot assess the genomic evolution

in later passage models, which are frequently used in the literature. Nevertheless, the on-going evolution of models suggests the value of generating banks of low passage PDX models and regular screening of PDX cohorts for acquired genomic changes.”

2.13. (V). The authors often referred to subclonal driver mutations, yet the definition of “driver mutations” in this context is unclear. For example, Supplementary Figure 13 shows many genes listed as “driver mutations” (e.g., BHMT2, ADCY2, MGA, RIMS2, TNR) yet the reason they are considered “drivers” is not provided. For example, BHMT2 is betaine-homocysteine S-methyltransferase 2 that plays critical role in methylation reactions. GeneCards mention that diseases associated with BHMT2 include paraneoplastic polyneuropathy and neural tube defects. Additional annotation as Census gene or Hallmark gene of BHMT2 is lacking from COSMIC. The reason for including it as a driver mutation in cancer including lung cancer needs clearer explanation or relevant citation.

Defining driver mutations in the context of cancer (and lung cancer specifically) remains a challenge for the field. To address this comment, we now use a more conservative definition which uses only those genes described as lung cancer-associated in Bailey et al. (2018), Berger et al. (2016), Martincorena et al. (2017) or through de novo dN/dS discovery in Frankell et al. (2023). This approach no longer calls *BHMT2* as a driver gene. We have expanded the Methods section to include a description of how we arrived at our driver mutation categorization as follows:

“For analysis of driver alterations, lung cancer-associated genes were derived from Bailey et al.⁶³, Berger et al.⁶⁴, Martincorena et al.⁶⁵ or through de novo dN/dS discovery in Frankell et al.³⁸. If the mutation was found to be deleterious (either a stop-gain or predicted deleterious in two out of the three computational approaches applied: Sift⁶⁶, Polyphen⁶⁷ and MutationTaster⁶⁸), and the gene was annotated as being recessive in COSMIC (tumor suppressor), the variant was classified as a driver mutation. Also, if the gene was annotated as being dominant (oncogene) in COSMIC, and we could identify ≥ 3 exact matches of the specific variant in COSMIC, we classified the mutation as a driver mutation, as per the approach in Frankell et al.³⁸”

Although we have de-emphasised this finding by moving it to Supplementary Figure 16 in the revised manuscript, the overall finding that driver mutation representation can differ between PDX models from the same primary tumor holds true; CRUK0995 R1 had a clonal *STK11* mutation that was entirely absent from the R3 PDX model, CRUK0606 R3 had a clonal *MGA* mutation that was absent from R6, and CRUK0816 R2 and R5 PDX models had a *CDKN2A* mutation that was absent from the R3 PDX model.

2.14. (VI). Additional issues:

- The sentence “..... than to spatially distinct tumor regions within the same tumor with the distance to non-region of origin comparable to that between primary tumor regions ...” (line 230-232) is difficult to understand.

Apologies. In the revised manuscript, we have re-worded this sentence. It now reads: “PDX models were significantly more similar to their region of origin than to other tumor regions from the same tumor (Figure 4C; $p = 9.3e-7$, Wilcoxon rank sum test), regardless of tumor histology (Supplementary Figure 15B). Indeed, the mutational distances between PDX models and regions of the tumor that were not selected to derive the PDX model (‘non-regions of origin’) were comparable to those between primary tumor regions (Figure 4C).”

2.15. - The term “homozygous” and “heterozygous” are used in the text (line 222-223) and as annotation of various colors in supplementary Figure 9A. However, their explanations are not provided in both the text and the figure legend, thus their meanings are unclear.

Based on the line and Figure references, we assume that this comment refers to the definition of homogeneous and heterogeneous regions rather than homo- and heterozygosity. Homogeneous regions are those that we infer consist of a single clonal cell population at the time of sampling, while heterogeneous regions consist of multiple detectable cell populations. We have now added a cartoon to Supplementary Figure 13A to show this, have revised the figure legend and now include these definitions within the text as follows:

“In the knowledge that primary tumor regions are heterogeneous (i.e. consist of multiple genomic subclones), we inferred the subclonal composition of P0 PDX models relative to their primary tumor region of origin. If multiple primary tumor subclones were found in the PDX model, we defined this as polyclonal engraftment, whereas if only a single primary tumor subclone was found in the PDX model, we defined this as monoclonal engraftment (Figure 4A). Of 42 unique P0 PDX models where WES data were available, 28 were monoclonal and 14 were polyclonal (Figure 4B). Where the clonality of the matched primary tumor region of origin could be inferred, three monoclonal PDX models arose from homogenous primary tumor regions (i.e. the PDX model was necessarily monoclonal due to the region of origin consisting of only a single clone). Of the 33 heterogeneous primary tumor regions, 13 gave rise to polyclonal PDX models, meaning that 20 PDX models from heterogeneous tumor regions were monoclonal (Supplementary Figure 15A).”

2.16. - In Supplementary Figure 10, black represents shared clusters. It is unclear why black is present in the phylogenetic trees but is not represented in the clone maps. The explanation of these plots in the figure legend is minimal. The figures may be understood better by genomic experts, but for most readers, more detail explanation in non-expert language should be provided.

We have added detail about the generation of clone maps to both the revised Figure legends and in the Methods section as follows:

Figure 4 legend: “Clonal composition is shown as clone maps where each hexagon corresponds to a clone from the phylogenetic tree that is present within the sample. Stacked clone maps (middle) are broken down as individual clones ordered by clone size (right).”

Figure 5 legend: “Regional clone maps (right) are a visual depiction of the P0 PDX sample composition and show the proportion of cells within the sample that can be attributed to a specific mutation cluster (as defined in the phylogenetic tree). Clusters indicated in black were shared clusters between primary tumor and PDX; gray - primary tumor specific clusters; colors (red, blue, green, purple, orange) indicate independent seeding clusters, and subsequent diversification in the PDX models is indicated by a gradient of each color (to white). Clusters highlighted with a bold black border were detected in the primary tumor while the other clusters are either PDX-specific or below the limit of detection in the primary tumor.”

Methods: “In Figures 4 and 5, the clonal composition of tumor samples are estimated using CONIPHER⁶², accounting for the nesting structure determined by the phylogenetic tree building. The images were generated using the cloneMap R package (version 1.0.0) available on github (<https://github.com/amf71/cloneMap>).”

2.17. - Line 644: the authors mentioned Copy number distance, they gave the

description, but did not provide the detail of calculation, how to get the copy number distance between two regions. Are they similar with mutation distance calculation?

In the revised manuscript, we have added a detailed Methods section describing our copy number distance metric as follows:

“The copy number distance gives an approximation of similarity between two regions relating to relative gains and losses of segments. If gains/losses of segments relative to ploidy are consistent across two regions the copy number distance is small; whereas when they diverge, e.g. a loss in one region and neutral copy number state in the other, the distance increases.

Given two regions i and j , and S being the total number of aberrant segments in either one or the other region, excluding all truncal copy number aberrations; the copy number distance is calculated as:

$$\frac{1}{\sum_{s=1}^S l_s} * \sum_{s=1}^S l_s * |CN_{i,s} - CN_{j,s}|$$

Where $CN_{i,s}$ and $CN_{j,s}$ are the total copy number of segment s in region i or j , respectively, and l_s is the length of segment s .”

2.18. - New NSG-adapted reference genome was concluded to decrease contaminating mouse reads when generating the cancer cell fraction (CCF) and phylogenetic trees and comparisons between primary region and PDX region, it is unclear how much the results would be significantly affected if only the standard murine genome (mm10) was used.

This is dependent on the analysis being performed. The results of high-level analyses in the manuscript would be unaffected by the inclusion or exclusion of this relatively small number of mutations. However, the issue becomes significant when considering selection, as the mutations are recurrent across patient samples. In our analyses of PDX-unique driver mutations, we find no such mutations that occur in multiple patients meaning that the mutation calls that are removed by use of the NSG-adapted reference genome are highly likely to be artefactual.

Reviewer #3: expertise in PDX evolution and bioinformatics

In this manuscript, Hynds et al. derived PDX models from multiple regions of primary NSCLC from patients enrolled in the TRACERx study, and explored the histological and genetic fidelity of these PDX models. They performed WES on the PDX models and compared PDXs to their primary tumors and to other PDXs derived from different regions of the same tumors. Using this approach, the authors tackled several questions: (1) The extent of genomic bottleneck upon engraftment; (2) The reproducibility of PDX derivation across spatially distinct samples from the same primary tumor; and (3) the genetic stability of PDX models over in vivo passaging. In addition, the manuscript reports an artifact in reads alignment and presents an NSG-adapted approach that improves the accuracy of sequencing analyses of PDX models.

This is an interesting study that expands the utility of the TRACERx project to address important questions related to PDX modeling. The experiments and the analyses are carried out well, and the manuscript is written clearly. The conclusions are thoughtful, and the paper will be of high value for those working with PDX models.

I therefore support its publication in Nature Communications. I do have a few suggestions that would strengthen the paper in my mind, and I'd encourage the authors to consider them:

We thank the reviewer for their kind words and helpful suggestions, which we have incorporated into the revised manuscript.

Major comments

3.1. The authors demonstrate the need for an NSG-adapted mouse reference genome. This is an important observation. I assume that the authors deposited the WGS data from the NSG mice in a publicly-available repository that can be downloadable without restrictions; please confirm. (a) Can the authors also provide a table with the ~7M SNPs that they identified to be different from the mm10 reference? Providing the code that spikes in the divergent NSG SNPs to the mm10 reference to improve filtering would also be very useful for the field. (b) Can the authors analyze previous large-scale studies (e.g., Sun et al Nat Commun 2021) to assess to what extent artifact mutations shown in Figure 3 affected previous analyses?

We agree that this aspect of the manuscript has good potential for re-use by others. In line with this, we have deposited the WGS raw data with the European Nucleotide Archive (ENA). This will be made publicly available upon final publication. We also provide the NSG-adapted mouse reference genome, the code used to generate it, and a VCF file containing all SNPs in a Zenodo repository. Accession numbers and/or web links for these resources are included within the revised manuscript. The Zenodo links are set up to allow reviewers to access the repositories.

It is difficult to assess the impact of the differences in mouse reference genomes in previous data, mostly because of the different pipelines that are employed in PDX model sequencing data processing and analysis. Some studies do not use a tool for removing mouse contamination, for example, while those that do often apply various filtering strategies that may partially mask this issue (for example, by limiting analyses to mutations present in the patient tumor, which would preclude investigation of PDX-specific mutations). That said, these mutations have been present in previous analyses, having been listed in Supplementary Tables associated with PMIDs 34215733, 32313009 and 28619968, which were PDX studies, and PMID 27693639, which involved sequencing of tumor cell cultures that were co-cultured with mouse embryonic feeder cells.

To assess the impact of using the mm10 reference genome in an independent NSG-based xenograft study, our collaborators at the University of Manchester processed and analyzed sequencing data from seven patient-derived xenografts that had whole-exome sequencing data available. Consistent with our findings, they found a reduction in mutations called using the NSG-adapted reference, which was reflected in a reduced number of shared mutations between samples (Supplementary Figure 12B). The specific *PPP1CC* mutation that we identified as the most common shared mutation in our analysis was identified in 4/7 xenografts. Thus, we confirm our findings using data from a center with different tumor sampling protocols, DNA extraction protocols, library preparation protocols, an independent sequencing facility and an independent data processing pipeline (GRCh38/hg38 rather than GRCh37/hg19; Mutect2 rather than VarScan2/Mutect1 with bespoke filtering) to those used in TRACERx.

3.2. The manuscript focuses almost entirely on point mutations. Given that (a) copy number alterations (CNAs) can be easily extracted from WES data; (b) that the Swanton lab has a long-standing interest and expertise in chromosomal instability and CNAs; and (c) that previous PDX analyses have focused on CNA evolution (refs

9-11) – this aspect of the paper should be further developed. Currently, the authors show in Supplementary Fig. 9 that there is high similarity between the mutation distance and the CNA distance of the PDXs and provide an example for CNA diversity across regions (and they also address CNAs in Supplementary Fig. 12). However, each of the Main Figures of the paper could benefit from an addition of a CNA-based analysis (alongside the mutation-based analysis). This would also allow for a direct comparison of the results to those from previous studies. For example, previous studies reported ~10%-20% CNA dissimilarity between primary tumors and PDXs, with high variance (both highly-similar and highly-divergent models). Is this the same range of changes observed in the current PDX cohort? (In this regard, it will be helpful to use additional measures for CNA comparison in addition to MSAI, such as the fraction of the genome that is CNA-concordant.)

Also, are these differences mostly the outcome of the subclonal CNA landscapes across tumor regions, as observed for mutations? Or would genomic evolution throughout propagation affect CNAs more than point mutations (e.g., due to potential negative selection against specific chromosome-scale alterations; or due to the p53 inactivation and chromosome instability of the PDXs)? All these questions can be addressed with the available data and are within the authors' expertise.

We agree that copy number analyses were underdeveloped in our first submission. We now include copy number distance in Figure 4E. We extended our investigation of PDX engraftment to include the proportion of the genome that is aberrant (Supplementary Figure 10A), the proportion of the aberrant genome that is subclonal (Supplementary Figure 10B) and whole-genome doubling (Supplementary Figure 10D). We find that LUAD tumors with a low proportion of the genome aberrant are less likely to engraft in PDX models. Since LUSC and 'other' histologies present comparable proportions of the genome aberrant to the LUAD tumors that engraft, and similar results were seen for wFLOH (Supplementary Figure 10C), this might suggest that there is a minimum threshold of genomic instability required for PDX engraftment that some LUAD tumors do not meet. In cross-genome plots (Supplementary Figure 9A), we investigate specific copy number changes that might be associated with PDX engraftment. In LUAD, we identified focal loss events that might be associated with PDX engraftment (Supplementary Figure 11A), including a segment on chromosome 9 that includes *CDKN2A*, *CDKN2B* and a cluster of type I interferon genes which was more likely to be subject to homozygous loss in engrafting regions. We also furthered our analyses of on-going evolution in PDX models to include analysis of the proportion of the genome that is aberrant (Figure 6D; Supplementary Figure 19A), LOH (Figure 6E; Supplementary Figure 19B), LOH reversion (Supplementary Figure 19C) and the proportion of the genome that is identical (Supplementary Figure 19F). We find that the proportion of the genome that is aberrant is lower in PDX samples than the tumor region of origin, consistent with a bottleneck event on engraftment. As expected the proportion of the genome subject to LOH increased through PDX engraftment and establishment.

3.3. When studying the genomic evolution of the PDXs, the authors compare primary-P3 samples and P0-P3 samples, reaching the conclusion that considerable genomic evolution can sometimes be observed, but most of the genomic divergence stems from the engraftment bottleneck (consistent with previous reports). It will be interesting to add a comparison of primary-P3 samples, to assess whether the genomic evolution of PDX models would make them diverge from the primary tumor throughout passaging. This is of course expected given the results that the clonal diversity decreases with passage, but it should be formally shown. (The distances between primary-Po and P0-P3 are not necessarily additive.)

Thanks. As suggested, we now include primary-P3 comparisons in Figure 6. As expected, we find that both the mutational (Figure 6B) and copy number (Figure 6C) distances

between the region of origin and P3 PDX samples is significantly higher than between the region of origin and P0 PDX samples, indicating an element of on-going divergence during passaging. While we don't see a difference in the proportion of the aberrant genome in this comparison (Figure 6D), we do see an increase in the proportion of the genome subject to LOH (Figure 6E).

3.4. Related to the previous point, the study compares P0 and P3 PDX samples. Many PDX cohorts comprise samples from much later passages, where genomic evolution might play a larger role. In case the authors have such matched late-passage PDXs available, it would be valuable to characterize those and subject them to the same types of analyses, as I'm not sure that P3 really represents a "late" passage. If no such examples are available, it's worth mentioning this point as a caveat in the Discussion.

We agree with the reviewer's concern about nomenclature for 'late passage' PDX models and have considered this carefully. In the revised manuscript, we now refer to 'initial' P0 and 'established' P3 PDX models to clarify the issue.

In the absence of WES for any PDX models at later the passage 3, we have added a statement as follows to the revised discussion to point this out: "A caveat of our study is that we did not analyze PDX tumors after P3, meaning that we cannot assess the genomic evolution in later passage models, which are frequently used in the literature."

REVIEWERS' COMMENTS

Reviewer #1 (Remarks to the Author):

The authors have improved the manuscript through the addition of new data, analyses, and explanation. I support publication in Nature Communications.

Reviewer #2 (Remarks to the Author):

Notwithstanding the limited sample size, the additional analyses and clarifications have significantly improved the manuscript and greatly appreciated. Overall, the revised discussion and conclusions are also reasonable and balanced.

Reviewer #3 (Remarks to the Author):

The authors have addressed my comments in a satisfactory manner, and I have no further comments. I recommend the acceptance of this manuscript for publication in nature communications.